



# Mapping the spatial distribution of NO₂ with in situ and remote sensing instruments during the Munich NO₂ imaging campaign

Gerrit Kuhlmann[1], Ka Lok Chan[2,3,*], Sebastian Donner[4], Ying Zhu[2], Marc Schwaerzel[1], Steffen Dörner[4], Jia Chen[5], Andreas Hueni[6], Duc Hai Nguyen[5,+], Alexander Damm[6,7], Annette Schütt[2], Florian Dietrich[5], Dominik Brunner[1], Cheng Liu[8], Brigitte Buchmann[1], Thomas Wagner[3], and Mark Wenig[2]

[1]Swiss Federal Laboratories for Materials Science and Technology (Empa), Dübendorf, Switzerland
[2]Meteorological Institute, Ludwig-Maximillians-University (LMU), Munich, Germany
[3]Remote Sensing Technology Institute (IMF), German Aerospace Center (DLR), Oberpfaffenhofen, Germany
[4]Max Planck Institute for Chemistry (MPIC), Mainz, Germany
[5]Environmental Sensing and Modeling, Technical University of Munich (TUM), Munich, Germany
[6]Department of Geography, University of Zurich (UZH), Zurich, Switzerland
[7]Swiss Federal Institute of Aquatic Science and Technology (Eawag), Dübendorf, Switzerland
[8]Department of Precision Machinery and Precision Instrumentation, University of Science and Technology of China (USTC), Hefei, China
*now at: Rutherford Appleton Laboratory Space, Harwell Oxford, United Kingdom
+now at: Leibniz Supercomputing Centre (LRZ), Garching bei München, Germany

**Correspondence:** Gerrit Kuhlmann (gerrit.kuhlmann@empa.ch)

**Abstract.** We present results from the Munich NO₂ imaging campaign (MuNIC) where nitrogen dioxide (NO₂) near-surface concentrations (NSC) and vertical column densities (VCD) were measured with stationary, mobile and airborne in situ and remote sensing instruments. The most intensive day of the campaign was 7 July 2016, when the NO₂ VCD field was mapped with the Airborne Prism Experiment (APEX) imaging spectrometer. The spatial distribution of APEX VCDs was rather smooth with a horizontal gradient between lower values upwind and higher values downwind of the city center. The NO₂ map had no pronounced source signatures except for the plumes of two combined heat and power plants (CHP). The APEX VCDs agree well with mobile MAX-DOAS observations from two vehicles conducted in the same afternoon (r = 0.55). In contrast to the VCDs, mobile NSC measurements revealed high spatial and temporal variability along the roads with highest values in congested areas and tunnels. The NOₓ emissions of the two CHP plants were estimated from the APEX observations using a mass-balance approach. The estimates are higher than reported emissions, but uncertainties are high because the campaign day was unstable and convective, resulting in low and highly variable wind speeds. The NOₓ emission estimates are consistent with CO₂ emissions determined from two ground-based FTIR instruments operated near one CHP plant. We conclude that airborne imaging spectrometers are well suited to map the spatial distribution of NO₂ VCDs over large areas. The emission plumes of point sources can be detected in the APEX observations, but accurate flow fields are essential to estimate emissions with sufficient accuracy. The application of airborne imaging spectrometers for studying NSCs, for example as input for epidemiological studies, is less straight forward and requires to account for the non-trivial relationship between VCDs and NSCs.



# 1 Introduction

Nitrogen oxides ($NO_x$ = NO + $NO_2$) are important precursors of ozone and particulate matter and thus play an important role in the formation of photochemical smog. Except close to source, $NO_2$ is usually the dominant component of $NO_x$ and is also the more critical species in terms of health effects (Beelen et al., 2014; Brunekreef and Holgate, 2002). Because of their negative effects on health, $NO_2$ concentration levels are limited by air pollution legislation, but these limits are still frequently exceeded in urban areas (European Environment Agency, 2019; World Health Organization, 2021). $NO_x$ is mainly emitted by

road traffic but also by residential heating, industrial facilities, power plants and some other combustion sources. Because of these localized emissions, $NO_2$ concentrations have a high spatial and temporal variability making high-resolution $NO_2$ maps an important tool for urban air pollution control and epidemiological studies (Maiheu et al., 2017).

$NO_2$ maps can be created in different ways each having its specific advantages and disadvantages: In situ ground measurements can be combined with geostatistical methods such as land-use-regression models to generate city-wide air pollution

maps (e.g., Mueller et al., 2015) but urban air quality monitoring networks are typically very sparse, limiting the accuracy of such maps. Dense networks of low-cost sensors have therefore been proposed as a complement to the traditional networks, but issues with precision, stability or specificity of existing sensors remain an obstacle for their widespread deployment (Heimann et al., 2015; Bigi et al., 2018; Karagulian et al., 2019). As an alternative, networks of ground-based remote sensing instruments including multi-axis differential optical absorption spectroscopy (MAX-DOAS) (Leigh et al., 2007) and tomographic

long-path (LP) DOAS systems (Platt et al., 2009) can be used to retrieve $NO_2$ maps. The disadvantage of this method is that a high spatial resolution can only be achieved with a very large number of light paths making a city-wide network expensive. Furthermore, the resulting $NO_2$ maps are mainly representing concentrations above buildings and not near the ground. A third option is the use of imaging spectrometers on satellites and aircraft. Current satellite instruments have spatial resolutions down to a few kilometers, which is still too coarse for resolving $NO_2$ in a city. However, satellite instruments have been shown to

be capable of observing the downwind plume of large cities (e.g., Beirle et al., 2011, 2019; Lorente et al., 2019). In contrast, airborne imaging spectrometers have inherently much higher spatial resolutions of a few tens of meters and thus can retrieve detailed $NO_2$ maps for whole cities (Heue et al., 2008; Lawrence et al., 2015; Popp et al., 2012; Schönhardt et al., 2015; Tack et al., 2017; Nowlan et al., 2016; Tack et al., 2019). However, satellite and airborne instruments measure vertical columns densities (VCD), while near-surface concentrations (NSC) is the quantity of interest for the assessment of air pollution expo-

sure. Since the linkage between VCDs and NSCs is variable and depends on various factors, algorithms have been developed to transfer VCDs to NSCs for satellite observations using statistical models trained with $NO_2$ monitoring networks (e.g. Xu et al.; de Hoogh et al., 2019; Kim et al., 2021).

These algorithms require large training datasets and cannot be applied to airborne measurement campaigns as typical urban monitoring networks are too small to train the relationship between VCDs and NSCs. This limits the current capacity to reliably

retrieve NSCs and to study the spatial variability of $NO_2$ and its sources in cities with airborne remote sensing observations. We





therefore conducted the Munich $NO_2$ Imaging Campaign (MuNIC) to validate airborne imaging spectrometers with ground-based observations and to advance the understanding on the relationship between VCDs and NSCs in Munich, Germany. We collected data with the Airborne Prism EXperiment (APEX) imaging spectrometer (Schaepman et al., 2015) and measured $NO_2$ VCDs and NSCs using mobile (MAX-DOAS, CAPS, CE-DOAS) and stationary instruments (LP-DOAS, MAX-DOAS)

as well as meteorological and other related parameters. We validate the APEX $NO_2$ map using the mobile MAX-DOAS observations, analysis the consistency of in situ and airborne $NO_2$ observations, study the relationship between VCDs and NSCs, and demonstrate the applicability of the collected data to estimate the emissions of the two largest point sources in the city.

## 2 Data and method

The MuNIC campaign was conducted from 1-13 July 2016 measuring $NO_2$ NSCs and VCDs with stationary, mobile and airborne instruments (Fig. 1; Tab. 1). The most intensive day of the campaign was the $7^{th}$ of July, when a map of $NO_2$ VCDs was retrieved with the APEX imaging spectrometer. On the same day, $NO_2$ NSCs and VCDs were measured with two vehicles from the Ludwig Maximilian University of Munich (LMU) and the Max Planck Institute for Chemistry (MPIC). Both vehicles were equipped with $NO_2$ in situ monitors and mobile MAX-DOAS instruments for measuring NSCs and VCDs, respectively.

In addition, stationary measurements were conducted at LMU's Meteorological Institute Munich (MIM) near the city center, at the Oscar-von-Miller (OvM) northeast of Munich, and near the combined heat and power (CHP) plant Munich South. In the following, the different measurements are described in more detail.

**Table 1.** Instruments and their measurements available during MuNIC on 7 July 2016.

| Instrument | Operators | Parameters | Measurement location | Measurement times (UTC) |
|---|---|---|---|---|
| APEX | Empa, UZH | $NO_2$ VCDs, HCRF | DLR aircraft | 12:16 - 12:56 |
| Mini MAX-DOAS | MPIC | $NO_2$ VCDs | LMU vehicle | 7:50 - 9:55, 11:30 - 15:30 |
| Tube MAX-DOAS | MPIC | $NO_2$ VCDs | MPIC vehicle | 8:00 - 9:45, 11:15 - 15:30 |
| CE-DOAS | LMU | $NO_2$ NSCs | MPIC vehicle | 8:10 - 9:40, 11:30 - 15:30 |
| CAPS | Empa | $NO_2$ NSCs | LMU vehicle | 7:50 - 9:55, 11:15 - 15:30 |
| MAX-DOAS (OvM) | LMU, USTC | $NO_2$ profiles | OvM tower | none |
| MAX-DOAS (MIM) | LMU | $NO_2$ profiles | MIM roof | 6:56 - 16:10 |
| LP-DOAS | LMU | $NO_2$ NSCs | MIM roof | all day |
| Bruker EM27/SUN FTIRs | TUM | $XCO_2$, $XCH_4$ | Munich South CHP plant | 9:20 - 16:10 |
| ASD | Empa, LMU, UZH | HCRF | hand held | 9:00 - 14:30 |



**Figure 1.** Map of Munich with overlayed APEX flight stripes (#1-4), routes taken by the LMU and MPIC vehicles in the afternoon of 7 July 2016 as well as locations of combined heat and power (CHP) plants, FTIR instruments, LfU monitoring sites, stationary MAX-DOAS instruments and the locations of surface reflectance measurements with the ASD instruments. Map data from © OpenStreetMap contributors 2021. Distributed under the Open Data Commons Open Database License (ODbL) v1.0.



## 2.1 Meteorological observations

Meteorological parameters were measured every minute at the MIM and the OvM tower at different altitudes. Air temperature,
air pressure, wind speed and direction, and global radiation are available. Wind speeds and directions were measured on the
MIM's rooftop at 30 m above ground. At the OvM tower, temperature and wind speeds were measured at 2, 5, 10, 20, 35 and
50 m above ground as well as wind directions at 10 and 50 m. Wind information is also available from measurements at the
Munich South CHP plant site at 15 m above ground and from the COSMO-1 and COSMO-7 model analysis product of the
Swiss Federal Office of Meteorology and Climatology (MeteoSwiss).

## 75  2.2 Monitoring stations

The Bavarian Landesamt für Umwelt (LfU) is operating a network of five monitoring stations in Munich (Figure 1) measuring
hourly concentrations of several air pollutants including $NO_2$ and meteorological parameters. The network consists of two
suburban background stations (Allach and Johanneskirchen), one urban background station (Lothstraße), and two urban road-
side stations at (Landshuter Allee and Stachus). $NO_2$ is measured with standard chemiluminescence $NO_x$ analyzers, which
use heated Molybdenum catalysts to convert $NO_2$ to NO before detection.

## 2.3 LP-DOAS

A long-path DOAS system was installed at the MIM's rooftop at 25 m above ground. The system measures the mean $NO_2$
concentrations along optical paths between the instrument and retro-reflectors installed at neighboring buildings using a blue
light-emitting diode (LED). The sampling time ranges from 30 to 90 s depending on visibility conditions. The measurement
setup and the $NO_2$ retrieval is described in detail by Zhu et al. (2020). Three optical paths were operated during the campaign
(Fig. S1 in the supplement): a 816 m path to the rooftop of the N5 building of the Technical University of Munich (TUM)
with the retro-reflector installed at 28 m above ground, a 2174 m path to the rooftop of the LMU physics building at 28 m
above ground, and a 3828 m path to the rooftop of the Hilton hotel building at 48 m height. The light paths cover the university
campus, a public park, residential areas and several roads.

## 90  2.4 Stationary MAX-DOAS

$NO_2$ profiles above roof level were measured with a MAX-DOAS instrument at the MIM's rooftop in the city center and a
second MAX-DOAS instrument on top of the OvM tower about 15 km north to the city center. Unfortunately, no measurements
are available at the OvM tower for 7 July 2016 due to technical issues of the instrument.

    The MAX-DOAS was programmed to measure scattered solar radiances at 9 elevation angles (2, 3, 4, 5, 6, 8, 15, 30 and 90°)
and 7 azimuth angles (0, 90, 135, 180, 225, 270 and 315°). In this study, only measurements at an azimuth angle of 0°(pointing
northwards) were analysed from which differential $NO_2$ slant columns (dSCDs) were retrieved using a DOAS analysis for a
wavelength range from 425 to 490 nm. The DOAS analysis considered $NO_2$, $O_4$, $O_3$ and $H_2O$ absorption cross sections, a



Ring spectrum as well as a polynomial of degree 5. Small shift and squeeze of the wavelength are allowed in the wavelength mapping process to compensate for small uncertainties caused by the instability of the spectrograph.

Aerosol extinction coefficient profiles were retrieved from the $O_4$ absorption bands at 477 nm using the Munich Multiple wavelength MAX-DOAS retrieval algorithm ($M^3$), described in detail in Chan et al. (2018, 2019, 2020). The algorithm uses the optimal estimation method (Rodgers, 2000) and the libRadtran radiative transfer code as the forward model (Mayer and Kylling, 2005; Emde et al., 2016). The aerosol profiles obtained from the procedure are used for the calculation of 1D-layer air mass factors (AMF) required for $NO_2$ profile inversion. The layer AMFs are calculated at 477 nm for the retrieval of $NO_2$

profiles using the Monte Carlo solver (MYSTIC) of libRadtran (Emde et al., 2016; Schwaerzel et al., 2020).

## 2.5   Airborne Prism Experiment (APEX)

APEX is a push-broom imaging spectrometer that has been developed for environmental monitoring (Schaepman et al., 2015). It measures radiance spectra with an optimized integration time simultaneously in 1000 across-track spatial pixels within a 28° field of view. At a flight altitude of 7360 m above ground, its spatial resolution is about 4 m × 6 m in across- and along-track

direction, respectively. For each pixel, spectral radiance is measured in the visible and near-infrared (VNIR, 372 - 1015 nm) and shortwave infrared (SWIR, 940 - 2540 nm) channel. To retrieve $NO_2$ VCDs, spectra were acquired in the unbinned mode providing the highest nominal spectral resolution of 0.86 nm full-width-at-half-maximum (FWHM) and 0.45 nm spectral sampling interval (SSI) in the VNIR channel with 334 spectral bands. Note that FWHM and SSI vary with wavelength and range from 0.86 to 15 nm (FWHM) and 0.45 to 7.5 nm (SSI), owing to the dispersion characteristics of the VNIR prism.

The APEX measurements were acquired along four pre-defined stripes shown in Fig. 1 on 7 July 2016 in the early afternoon (14:16 - 14:56 CEST). The four stripes cover a large fraction of the city with a large overlap between the stripes to increase the amount of APEX observations in the city center where most ground-based observations were conducted. In addition, the stripes cover forest and agriculture land that were used as background spectra in the DOAS retrieval.

APEX $NO_2$ columns were retrieved with the second version of the Empa APEX $NO_2$ retrieval algorithm (Popp et al., 2012,

for Version 1). To increase the signal-to-noise ratio, 20 × 10 APEX pixels were spatially binned in across- and along-track direction, respectively. As a result, the spatial resolution of the APEX instrument is reduced to about 80 m × 60 m. An in-flight spectral calibration was conducted to improve the accuracy of center wavelength positions and the full-width-at-half-maximum (FWHM) of the instrument's slit function (Kuhlmann et al., 2016). This was required because the default spectral calibration provided by the APEX processing and archiving facility (Hueni et al., 2009, 2013) was not sufficiently accurate for the retrieval

of $NO_2$.

$NO_2$ dSCDs were retrieved from the spatially binned APEX spectra using the flexDOAS Python library (Kuhlmann, 2021b). The last 50 spectra at the western end of each stripe were used as reference spectrum for each across-track position. The DOAS analysis was applied to a window from 470 to 510 nm where we fitted $NO_2$ and $O_4$ absorption cross sections, a Ring spectrum, a 5-degree polynomial as well as a relative offset fitted as a quadratic polynomial that was subtracted both from the measurement

and reference spectrum.





The $NO_2$ dSCDs are converted to VCDs using AMF:

$$VCD = \frac{dSCD + VCD_{ref} \cdot AMF_{ref}}{AMF} \tag{1}$$

where $VCD_{ref}$ and $AMF_{ref}$ are the reference VCDs and AMFs for each across-track position. Mobile MAX-DOAS observations conducted in the reference area during the APEX flight were used as reference VCDs ($VCD_{ref}$).

To calcuate the AMFs, 1D-layer AMFs were computed with the MYSTIC solver of the libRadtran model (Emde et al., 2016; Schwaerzel et al., 2020) depending on sun position, instrument viewing direction, surface reflectance, surface elevation and atmospheric scattering by molecules and aerosols using an US standard atmosphere. Surface reflectances were taken from the APEX surface reflectance product at $490\,nm$ (c.f. Sec. 2.8) in the center of the DOAS fitting window assuming Lambertian equivalent reflectance (LER). Aerosol scattering was included using an aerosol optical depth of 0.10 measured by

the AERONET station at the MIM's rooftop and libRadtran's default aerosol profile. Total AMFs were computed from the 1D-layer AMFs using the $NO_2$ profile measured by the MAX-DOAS instrument at the MIM's rooftop.

The APEX $NO_2$ VCDs were destriped to account for small variations of dSCDs and true $VCD_{ref}$ in across-track direction by subtracting deviations from a cubic polynomial fitted to the mean across-track VCDs. In addition, we performed a bias-correction between the different aircraft overpasses by adding a constant offset to each stripe so that the mean values of the

overlapping parts of two neighbouring stripes are identical. Both the across-track variations as well as the difference between stripes are mainly caused by differences in spectral and radiometric calibration (Kuhlmann et al., 2016). The $NO_2$ VCDs were then mapped on a longitude-latitude grid with about $10\,m$ resolution for comparison with the ground-based observations using the gridding algorithm of Kuhlmann et al. (2014) (Code repository: Kuhlmann, 2021c).

The uncertainties in $NO_2$ VCDs were calculated as the quadratic sum of the uncertainties in dSCDs, AMFs, and $SCD_{ref}$:

$$\sigma_{VCD} = \sqrt{\left(\frac{\sigma_{dSCD}}{AMF}\right)^2 + \left(\frac{\sigma_{SCD_{ref}}}{AMF}\right)^2 + \sigma_{AMF}^2 \left(\frac{VCD}{AMF}\right)^2} \tag{2}$$

where $\sigma_{dSCD}$ was estimated from the DOAS analysis. The uncertainties in AMFs and $SCD_{ref}$ were estimated from the comparison with the ground-based remote sensing observations conducted during the campaign.

## 2.6   Mobile measurements

Mobile measurements were conducted with two vehicles operated by MPIC and LMU, respectively. Each vehicle was equipped

with an in situ instrument measuring near-surface concentrations (CE-DOAS and CAPS) and a spectrometer measuring $NO_2$ column densities (Tube and Mini MAX-DOAS) (see Table1). The measurements were conducted along varying routes in the city. The MPIC vehicle mostly drove circles in the city center to capture the $NO_2$ fields in across-track direction of the APEX measurements, while the LMU vehicle was driving from the southwest to the northeast of the city along the highway or on smaller roads in the city center to sample in the along-track direction of APEX.


### 2.6.1 Mobile in situ instruments

Near-surface $NO_2$ concentrations were measured with two highly sensitive and specific in situ instruments: A T500U CAPS $NO_2$ Analyzer from Teledyne API that is routinely used in the Swiss National Air Pollution Monitoring Network (NABEL). CAPS consists of a blue LED, a measurement chamber with two highly reflective mirrors and a vacuum photodiode detector. The instrument uses the Cavity Attenuated Phase Shift (CAPS) technique to directly measure $NO_2$ (Kebabian et al., 2005, 2008). The CAPS was installed in the LMU vehicle with the sample inlet fixed at the roof of the vehicle at about 2 m height. The $NO_2$ concentrations were recorded every 2 s.

The second instrument was the Broadband Cavity Enhanced Differential Optical Absorption Spectroscopy (CE-DOAS), which uses a blue LED, a measurement chamber and a spectrometer to obtain $NO_2$ concentrations using the DOAS technique between 435.6 and 455.1 nm (Platt et al., 2009; Zhu et al., 2020). The CE-DOAS was installed in the MPIC vehicle with the inlet located at the front right window of the vehicle at 1.5 m above the ground and measurements were recorded every 2 s.

CAPS and CE-DOAS agreed well when operated together on the LMU vehicle on 1 and 4 July 2016. Pearson correlation coefficients were high with 0.995 and 0.984, but it was necessary to shift CAPS measurements by 16.9 s and 5.6 s, respectively (Fig S2). The time differences were caused by non-synchronised computer clocks and instrument differences such as length of tubing and integration times. On campaign day, GPS times were used to minimize time differences between instruments.

### 2.6.2 Mobile MAX-DOAS measurements

The mobile MAX-DOAS measurements were carried out using two instruments operated by MPIC: The Tube MAX-DOAS, which was mounted on the MPIC vehicle, is a scientific grade instrument with a high signal-to-noise ratio and stable spectroscopic properties (see e.g., Kreher et al., 2020). The Mini MAX-DOAS is a more compact instrument but with lower signal-to-noise ratio and less stable spectral properties (see e.g., Shaiganfar et al., 2011). This instrument was mounted on the LMU vehicle. Because of the different signal-to-noise ratios, the integration times for individual measurements were set to 30 s for the Tube MAX-DOAS and 60 s for the Mini-MAX-DOAS instruments, respectively. One complete elevation sequence contained 6 (Tube MAX-DOAS) or 7 (Mini MAX-DOAS) measurements at low (22°) elevation angle, followed by one measurement in zenith direction. The line of sight of the Mini MAX-DOAS instrument was in driving direction and the one of the Tube MAX-DOAS instrument was directed backwards with respect to the driving direction. $NO_2$ was analysed in the spectral range from 400 to 439 nm using a fixed daily Fraunhofer reference measured in zenith direction. Only results with root mean square (RMS) values below $8\times10^{-4}$ (Tube MAX-DOAS) and below $2\times10^{-3}$ (Mini MAX-DOAS) were considered for further processing. The method described in Wagner et al. (2010) and Ibrahim et al. (2010) was applied to determine the $NO_2$ absorption in the Fraunhofer reference spectrum and to correct for the changing stratospheric $NO_2$ absorption.

### 2.7 Fourier transform infrared (FTIR) spectrometers

Two ground-based Fourier transform infrared (FTIR) spectrometers were deployed near the Munich South CHP plant to measure column-averaged dry air mole fractions of $CO_2$ and $CH_4$ (XCO$_2$ and XCH$_4$). The instruments are owned by KIT and



TUM and are identified as EM27 KIT and EM27 TUM, respectively. Both instruments are operated by TUM. The compact and mobile solar-tracking FTIR spectrometers (Bruker EM27/SUN, Gisi et al. (2012)) measure spectra in the wavenumber range from 6000 to 9000 $cm^{-1}$. By placing one spectrometer upwind and the other downwind of the CHP plant, differential column

measurements (DCM) can be used to determine the $CO_2$ and $CH_4$ emissions from this source. DCM have proven to be an effective method for determining emissions from different types of sources (Hase et al., 2015; Chen et al., 2016; Dietrich et al., 2021; Toja-Silva et al., 2017; Zhao et al., 2019). Differential column measurements exhibit a very high precision of 0.01% for $XCO_2$ and $XCH_4$ at 10-min. integration time (Chen et al., 2016; Hedelius et al., 2016).

To capture the emission plume, the spectrometers were placed based on the forecasted wind direction. On 7 July 2016,

the wind was blowing from the northwest in the morning turning to northeast in the afternoon. One spectrometer was placed southwest of the CHP plant to capture the plume in the morning and the other spectrometer south to capture the plume at noon. As a result, the stations acted alternately as downwind sites and as background sites throughout the day.

### 2.8 Surface reflectance

Surface reflectance is a critical input parameter for the AMF computation, because uncertainties in the reflectance can have

a strong impact on the $NO_2$ VCDs. The APEX reflectance product is calculated using the atmospheric correction software ATCOR-4 (R.Richter and Schläpfer, 2002). In short, atmospheric water vapour and aerosol optical depth are estimated from APEX radiance data, which are used, together with other parameters describing the sun-observer geometry, to obtain representative atmospheric transfer functions (e.g. transmittance, spherical albedo, path radiance). The transfer functions were pre-calculated with MODTRAN-5 (Berk et al., 2005) and stored in a look-up table. Radiance data are also used to estimate

spectral-non-uniformities (e.g., spectral shift, band broadening). This information is essential to spectrally convolve identified atmospheric transfer functions and eventually retrieve surface hemispheric-conical reflectance factors (HCRF).

To evaluate the APEX surface reflectance product, HCRF spectra were collected with an hand-held Analytical Spectral Device (ASD) field spectroradiometer during the campaign. In total, 14 spectra were measured on 7 July 2016 from 11:05 to 16:19 CEST. All ASD measurements were located in the second and third APEX stripe and included surfaces of varying type

and brightness. The locations are shown in Figure 1 and details are listed in Table S1 in the supplement.

## 3 Results

### 3.1 General situation

The APEX flight window was part of a larger measurement campaign in June and July 2016 with various targets. The choice of a suitable day for Munich based mainly on flight permission and weather conditions. The first two weeks of July 2016

were mostly cloudy with daily mean temperatures and wind speeds ranging from 12–27 °C and 1.0–3.9 $m\,s^{-1}$, respectively. A favorable situation with almost no clouds was predicted for 7 July 2016 for the area of Munich, which allowed to conduct the APEX measurements in the early afternoon (14:16 - 14:56 CEST).





Figure S3 shows the time series of meteorological parameters (temperature, wind speed and direction and global radiation) measured at MIM and at the OvM tower on this day. In the morning, small cumulus clouds were still present over Munich, which can be seen as a reduction in global radiation. In the afternoon, the clouds mostly vanished making it possible to conduct the APEX flight. Figure S4 shows the true color composite of the APEX measurements showing only a small cloud at the edges of the image.

Wind speeds were low during night (about $1\,\mathrm{m\,s^{-1}}$) and slightly higher and highly variable during the day (1 to $6\,\mathrm{m\,s^{-1}}$). The wind directions were mostly northwesterly during day turning to northeasterly in the evening. We estimated the dispersion category during daytime as very unstable (category V) based on the procedure published by the Association of German Engineers, which considers mainly wind speed, cloud fraction and hour of day (VDI - Fachbereich Umweltmeteorologie, 2009, Annex A).

Figure 2a shows the time series of $NO_2$ NSCs at the monitoring stations on 7 July 2016. The $NO_2$ measurements show a clear diurnal cycle with a morning peak during rush hour and an increase in the evening when the boundary layer becomes stable again and the sources (traffic) are still active. The APEX flight was performed around the time of lowest NSCs. Concentrations were about 5 ppbv at the two suburban background stations (Allach and Johanneskirchen) and somewhat higher with 11 ppbv at the urban background station (Lothstrasse). Different from these stations, the two urban traffic stations showed only little variations during the day due to their proximity to emission sources. Concentrations during the APEX flight were 35 and 66 ppbv at Stachus and Lanshuter Allee, respectively. The LP-DOAS also measured the diurnal cycle with molar fractions similar to the suburban background stations due to the elevated light paths measuring $NO_2$ above the urban canopy. The LP-DOAS measured $NO_2$ concentrations of about 5 ppbv during the APEX overpass.

The MAX-DOAS instrument on the rooftop (Fig. 2c) measured highest $NO_2$ VCDs in the early morning with up to $504\,\mathrm{\mu mol\,m^{-2}}$. $NO_2$ VCDs dropped substantially to about $104\pm15\,\mathrm{\mu mol\,m^{-2}}$ in the afternoon. Note that $100\,\mathrm{\mu mol\,m^{-2}}$ are about $6\times10^{15}$ molecules $\mathrm{cm^{-2}}$. Since the temporal variability of $NO_2$ VCDs was small in the afternoon, differences between mobile and APEX measurements due to different measurement times are likely small.

Figure 2d shows averaged $NO_2$ profiles retrieved in the morning and in the afternoon. The profiles were retrieved for an azimuth angle of 0 degrees, i.e. looking northwards. The mole fractions in the lowest layer (0 - 200 m) were only 4.6 and 2.0 ppbv in the morning and afternoon, respectively, which is smaller than the 8.4 and 3.8 ppbv measured by the LP-DOAS system on average.

## 3.2 Mobile in situ measurements

Figure 3a shows the $NO_2$ mole fractions along the routes taken by the LMU and MPIC vehicles on 7 July 2016 in the afternoon. The markers for the LfU sites and lines for LP-DOAS show mean values in the afternoon (11:00 - 17:00 UTC). Figure 3b and c show the corresponding time series of $NO_2$ mole fractions. A corresponding figure showing measurements during the morning (7:00 - 10:00 UTC) is available in the supplement (Fig. S5).

$NO_2$ mole fractions varied strongly along the route ranging from 0 to 890 ppb with highest values being observed in congested areas such as in front of traffic light or in tunnels. However, while the two-second measurements show strong variability



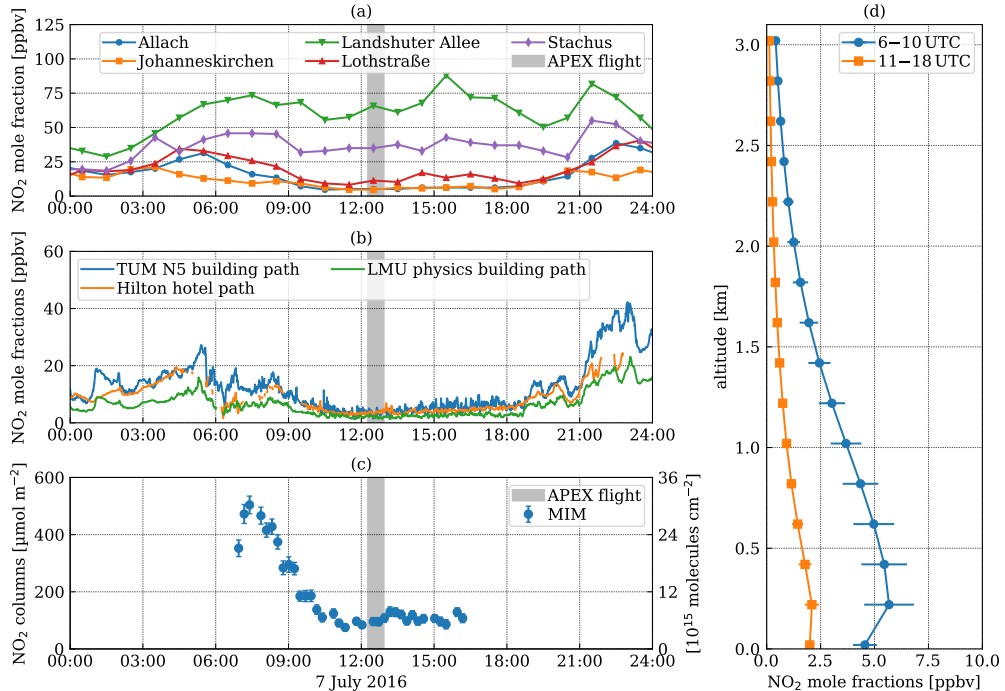

**Figure 2.** (a,b) Time series of $NO_2$ mole fractions from monitoring stations and the LP-DOAS systems. (c) Time series of vertical column densities and (d) mean $NO_2$ profiles from the MAX-DOAS instrument on the MIM's rooftop on 7 July 2016.

with very high values, 30-minute averages are similar to hourly measurements at the two roadside stations Stachus and Landshuter Allee.

## 3.3 APEX $NO_2$ observations

### 3.3.1 APEX $NO_2$ retrieval algorithm

The true color images of the four APEX stripes are shown in Fig. S4 in the supplement. The stripes were mostly cloud-free with only a small cumulus cloud in the western edge of the second stripe.

The spectral calibration was much more stable than on other flights investigated by Kuhlmann et al. (2016) likely because the measurements were conducted after the long transfer flight from Zurich to Munich prior to the measurements, allowing the system to reach a stable pressure and temperature state. The in-flight spectral calibration shows the known spectral smile of the APEX instrument in across-track direction (Fig. S6a) but no strong drift during data acquisition in along-track direction (Fig. S6b). The instrument slit function is about 30% wider in-flight than the laboratory calibration, which is a known but not fully understood characteristic of the APEX instrument that is likely related to stray-light and vignetting (Kuhlmann et al., 2016).





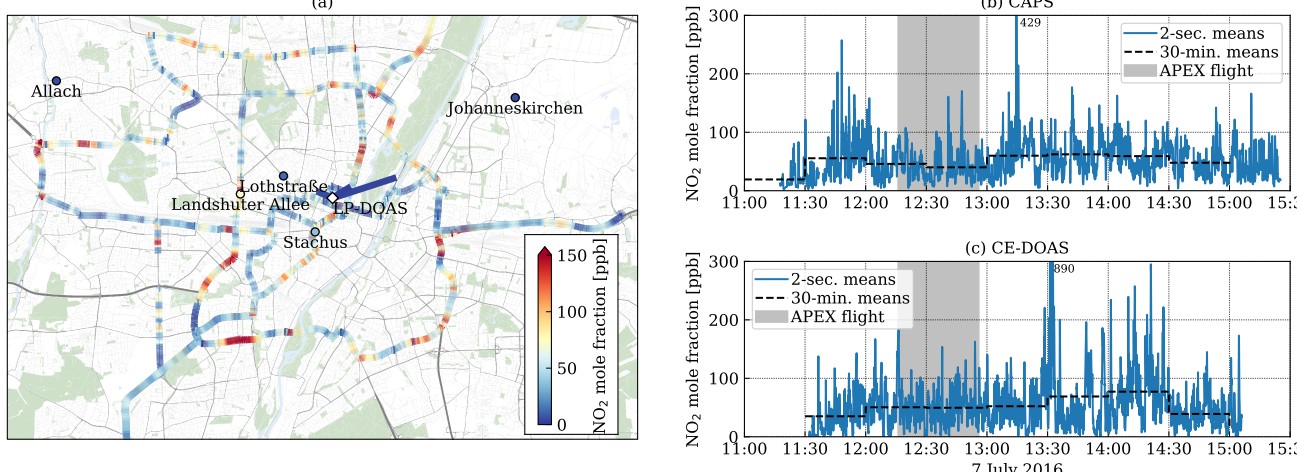

**Figure 3.** (a) Map of $NO_2$ mole fractions measured by LMU and MPIC vehicles in the afternoon. Locations of the LfU monitoring stations are shown as circles colored by the mean afternoon $NO_2$ mole fraction. (b) and (c) show the time series of $NO_2$ mole fractions measured by the CAPS and CE-DOAS instrument on the LMU and MPIC vehicle, respectively. Map data from © OpenStreetMap contributors 2021. Distributed under the Open Data Commons Open Database License (ODbL) v1.0.

Figure S7 shows an example of the DOAS analysis for an APEX pixel acquired inside the plume of the Munich South CHP plant (stripe: 2, across-track: 42, along-track: 314) with a root mean square (RMS) of $1.22 \times 10^{-3}$ and an $NO_2$ dSCD of $621 \pm 130\,\mu\text{mol m}^{-2}$.

    The distributions of RMS, dSCDs, HCRFs, AMFs and VCDs are shown in Fig. 4 for the second stripe. The distributions for the other APEX stripes are shown in the supplement (Fig. S8-S10).

For all stripes, RMSs do not vary strongly spatially ranging from 0.49 to $1.00 \times 10^{-3}$ (5th to 95th percentile) with a mean value of $0.73 \times 10^{-3}$. The values are smallest in the area used for computing the reference spectra (Fig. 4a). RMS are highest over very bright surfaces and increase with distance from the reference area. This dependency on along-track position is likely caused by a mismatch in spectral calibrations between reference and measurement spectrum that increases with distance from the reference area. RMSs also vary in across-track direction where positions have higher RMS than other also likely caused by
small differences in the spectral calibration.

    The $NO_2$ dSCDs show no clear spatial pattern, but values are highest in the middle over the city and lowest at the right side over the forest used as reference area (Fig. 4b). The average dSCDs is $178\,\mu\text{mol m}^{-2}$ but values range between -71 and $417\,\mu\text{mol m}^{-2}$ (5th to 95th percentile).

    HCRFs at 490 nm in the APEX product varied over the city from 0.02 to 0.12 (5th to 95th percentile) with a mean of 0.06.
The HCRFs were smaller with an average value of 0.03 (range: 0.01–0.06) over the forest in the west of the city, which was used as a reference area for the DOAS analysis. The AMFs depend mainly on HCRFs and range between 1.1 and 1.9 with an average of 1.5 (Fig. 4d).

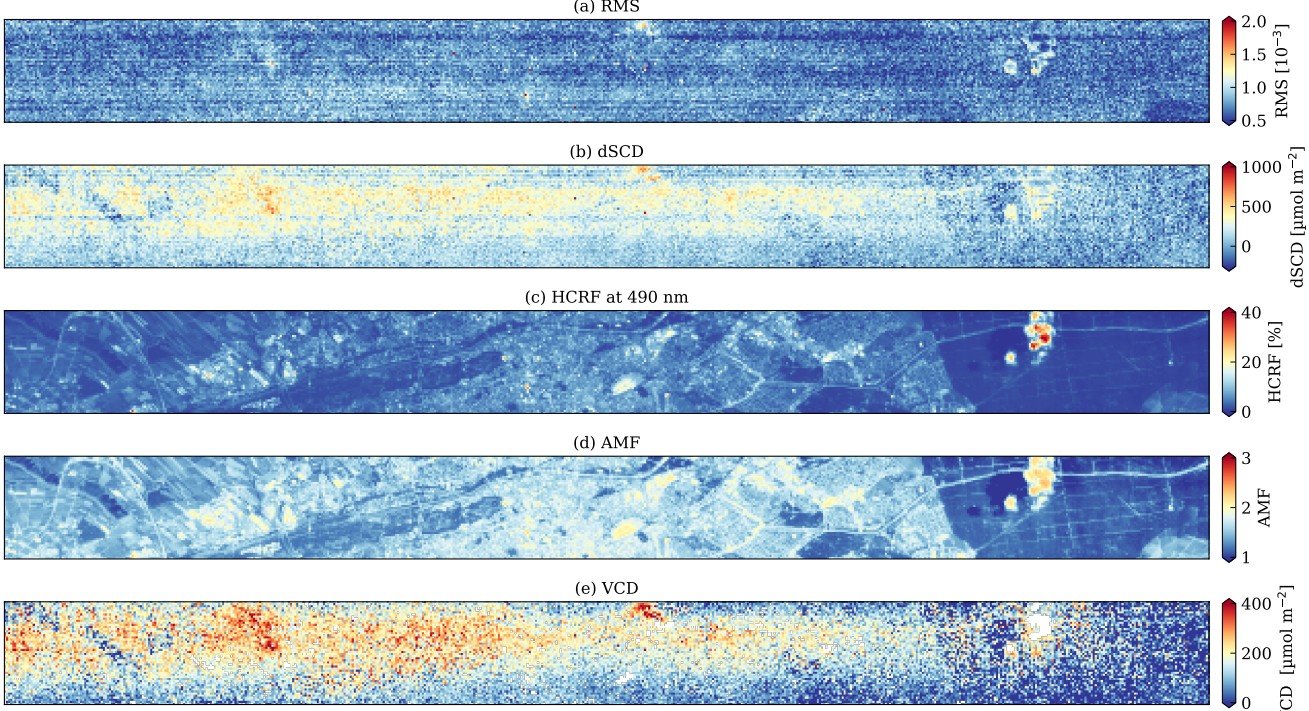

**Figure 4.** (a) Root mean square (RMS), (b) $NO_2$ differential slant column densities (dSCD), (c) hemispheric-conical reflectance factors (HCRF), (d) air mass factors (AMF) and (e) $NO_2$ vertical column densities (VCD) of the second APEX stripe.

The $NO_2$ VCDs computed from dSCDs and VCDs (Eq. 1) have an average of $115\,\mu mol\,m^{-2}$ and range from -37 to $294\,\mu mol\,m^{-2}$ ($5^{th}$ to $95^{th}$ percentile). The destriping algorithm successfully removes the stripes visible in the dSCDs (Fig. 4e).

The destriped VCD field shows clearer features such as two enhancements in the left and centre of the stripe, which match the locations of the Munich North and Munich South CHP plants.

### 3.3.2 APEX $NO_2$ uncertainties

The uncertainties of the VCDs were obtained from the uncertainties of dSCDs, $SCD_{ref}$ and AMFs (Eq. 2). The dSCD uncertainty was obtained directly from DOAS fitting routine and varied between 18 and $185\,\mu mol\,m^{-2}$ with $56\,\mu mol\,m^{-2}$ on

average.

The uncertainties in the AMFs are caused mainly by uncertainties in the surface reflectance product and the a priori $NO_2$ profile used in AMF calculations. To estimate the uncertainty of the APEX surface reflectance product, we validated the product with the 14 ASD measurements in the city center covered by the second and third APEX stripe (Fig. 1 and S11, and Tab. S1). Figure 5 shows two examples of HCRF spectra over a dark and bright surface. APEX and ASD HCRF agree very well at

490 nm with a correlation coefficient of 0.99, but the APEX product tends to underestimate high reflectances (slope: 1.21, intercept: -0.02). The root mean square difference (RMSD) between APEX and ASD HCRFs is 0.014 considering only ASD





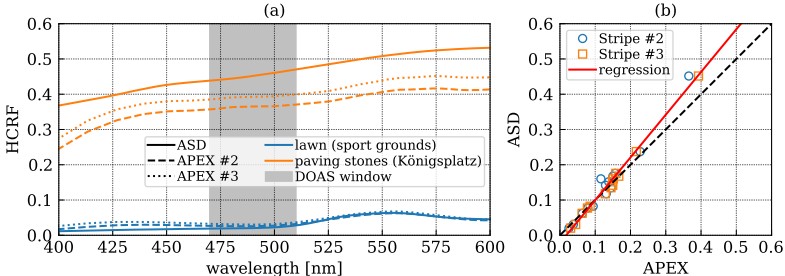

**Figure 5.** (a) APEX and ASD HCRF spectra for a dark and bright surface. (b) Scatter plot comparing APEX and ASD at 490 nm over all 14 targets. The regression line has a slope of 1.211 and an intercept of -0.022. The correlation coefficient is 0.99.

values smaller than 0.30, which is a relative uncertainty of about 23% for the mean HCRF over the city (HCRF = 0.06). We computed the $1\sigma$ uncertainties of the AMFs using an uncertainty of 0.014 in HCRFs, which results in AMF uncertainties of 0.15 and 0.08 for HCRFs of 0.03 and 0.06, respectively.

The uncertainties in the a priori $NO_2$ profiles translate to about 10% uncertainty in AMFs for satellite $NO_2$ products (e.g., Boersma et al., 2011). In our study, the a priori $NO_2$ profile was obtained from the MAX-DOAS instrument on the MIM rooftop, which should provide a representative profile for the city above building level. Since the instrument is not sensitive to near-surface $NO_2$, we analyzed the impact of the $NO_2$ profile sensitivity by replacing the mole fraction in the lowest layer (0-200 m) with values ranging from 0 to 100 ppbv. We find that the corresponding AMF uncertainty is smaller than 0.10 for

LERs larger than 0.20 and up to 0.16 for a black surface (LER = 0.00).

The total AMF uncertainties are obtained as sum of variances of the individual components (surface reflectrance and a priori $NO_2$ profile). We calculated that AMF uncertainties decrease from 0.40 to 0.10 for LERs increasing from 0.00 to 0.05 and is constant at 0.10 for LERs larger than 0.05. Uncertainties might be somewhat larger, because uncertainties in the aerosol profiles are not taken into account.

The uncertainty of the reference slant column density ($SCD_{ref}$) was estimated to be about 25% considering the uncertainty of the VCDs retrieved from the Mini MAX-DOAS ($\sigma_{SCD} < 15\%$) and the uncertainty of the AMFs averaged over the forest used as reference area ($\sigma_{AMF} \approx 0.13$).

The VCD uncertainties were computed from Eq. (2). The uncertainty is dominated by the dSCD uncertainty that accounts for 87% of total uncertainty due to the low signal-to-noise ratio of the APEX measurements. The second most important term is the

AMF uncertainty accounting for 12% of the total uncertainty while the uncertainty of the reference SCD is negligible. Figure 6a shows the estimated uncertainties for the second stripe. The uncertainties are highest over dark surfaces (forest, parks and water surfaces), because the uncertainty inversely depends on AMFs (Eq. 2). The uncertainty ranges from 34 to 86 $\mu$mol m$^{-2}$ (5th-95th percentile) with an average of 56 $\mu$mol m$^{-2}$ (Fig. 6b). Since the uncertainty depends on VCD, uncertainties slightly increase with VCDs (Fig. 6c).





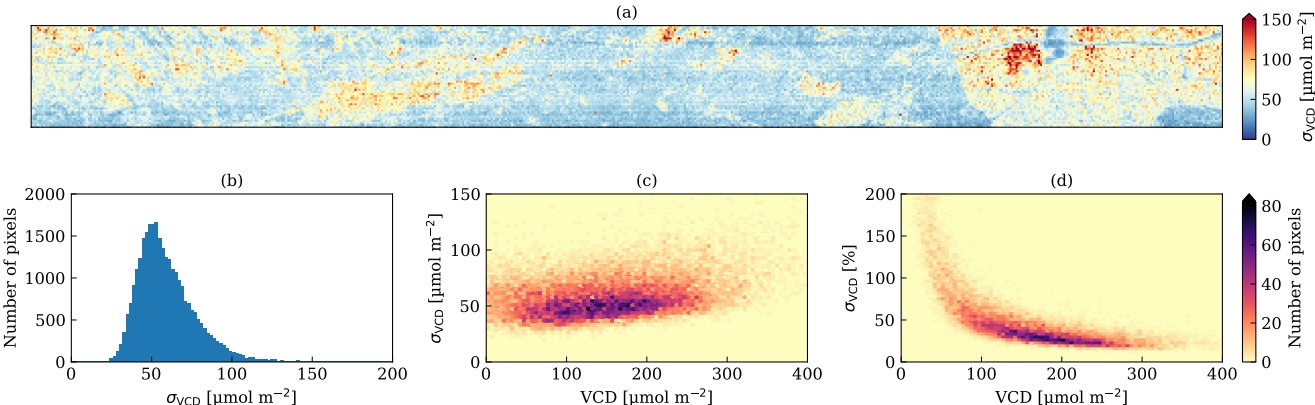

**Figure 6.** Estimated uncertainty of $NO_2$ VCDs for the second APEX stripe showing (a) the spatial distribution, (b) the distribution of uncertainties and the dependency on the $NO_2$ VCDs of the (c) absolute and (d) relative uncertainty.

### 3.3.3 APEX $NO_2$ map

Figure 7a shows the map of $NO_2$ VCDs from the APEX instrument. A Gaussian smoothing filter ($\sigma \approx 50$ m) was applied to reduce spatial noise. The map shows lower values in the northwest upwind and higher values in the southeast downwind of the city center. The locations of the Munich North and Munich South CHP plants are marked by two orange triangles. The emission plume of both CHP plants are clearly visible in the APEX data. Otherwise, no small-scale structures, such as roads, can be identified in the map. This is consistent with previous studies where only the overall city plume and elevated plumes of larger point sources were discernible (e.g., Popp et al., 2012; Tack et al., 2017, 2019). Recent model studies demonstrate that this is at least partially explainable by 3D radiative transfer effects (Schwaerzel et al., 2021), while, in addition, atmospheric mixing is expected to play a role especially in the afternoon when turbulent mixing is highest.

The APEX $NO_2$ product also shows artefacts near the stripe edges that are likely related to insufficient knowledge about the spectral calibration and vignetting that affect the accuracy of the instrument slit function (Kuhlmann et al., 2016). The $NO_2$ map also shows unrealistic low values over water surfaces that also have been noticed by Tack et al. (2017).

### 3.4 Comparison of APEX and MAX-DOAS observations

Figures 7b and c show the time series of spatially but not temporally collocated APEX and MAX-DOAS $NO_2$ VCDs, which were obtained by averaging the APEX values along the vehicle paths $\pm 30$ s and $\pm 15$ s for the Mini- and Tube-MAX DOAS, respectively. The LMU vehicle drove along the magenta line (Fig. 7a) leaving the urban area in the southeast and northwest. The time series shows local $NO_2$ minima when the vehicle was at the southwestern border of the APEX map at 12:20 and 14:35 UTC and at the northwestern border of the APEX map at 13:25 UTC. The MPIC vehicle was driving in anticlockwise direction around the city center along the red line in Fig. 7a. $NO_2$ VCDs were highest when the vehicle was located in the eastern part of the city (12:30 and 14:30 UTC). A map showing the time labels is provided in the supplement (Fig. S12).

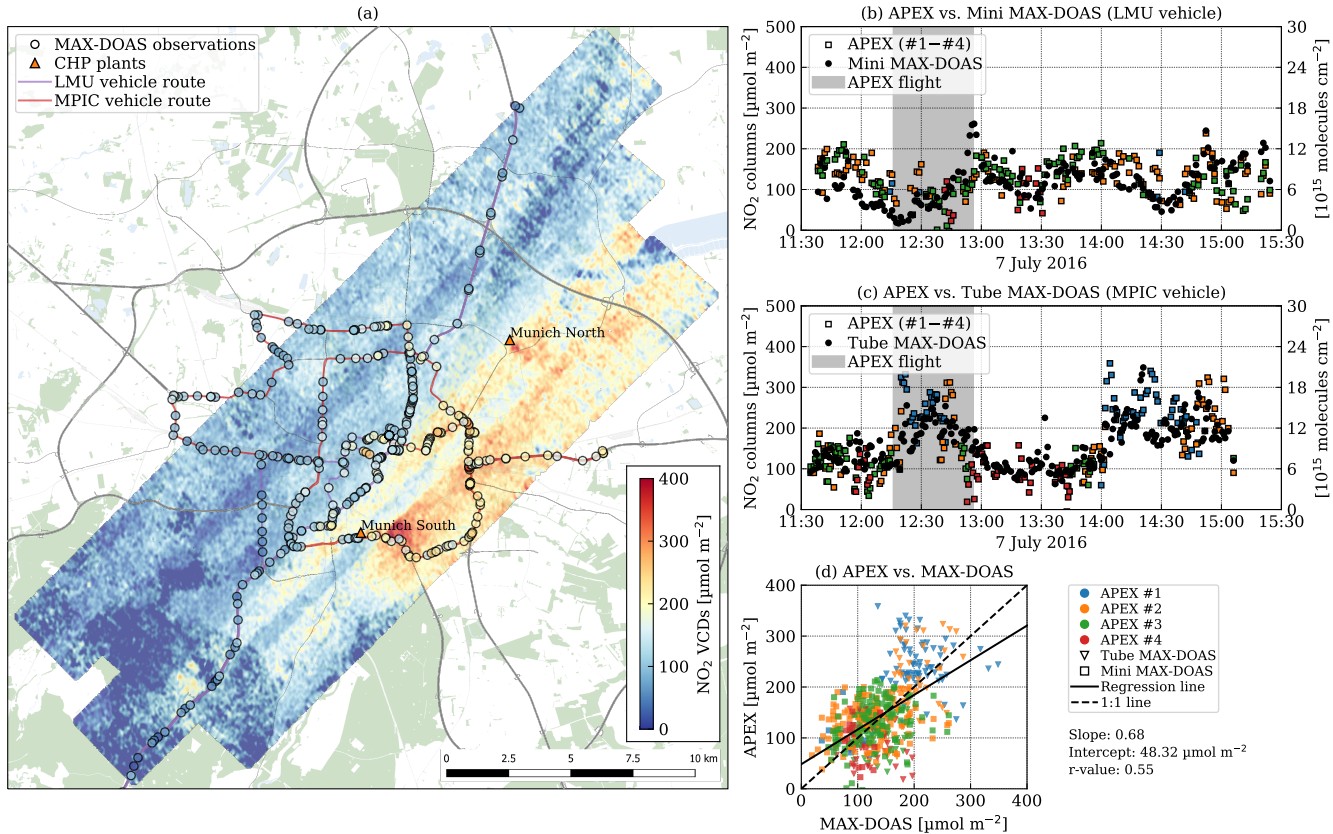

**Figure 7.** (a) Map $NO_2$ VCDs from APEX and mobile MAX-DOAS measurements on 7 July 2016 (afternoon). (b,c) Time series of spatially co-located APEX and (b) Mini MAX-DOAS and (c) Tube MAX-DOAS VCDs. (d) Scatter plot showing MAX-DOAS and APEX $NO_2$ VCDs for spatially co-located observations. Map data from © OpenStreetMap contributors 2021. Distributed under the Open Data Commons Open Database License (ODbL) v1.0.

In total, 518 co-located APEX and MAX-DOAS observations are available. APEX and MAX-DOAS $NO_2$ VCDs agree quite well with a moderate correlation coefficient $r$ of 0.55. The regression line has a slope of 0.68 and an intercept of $48.3\,\mu\mathrm{mol\,m^{-2}}$. The discrepancy between airborne and ground-based observations is largely explainable by the relatively high uncertainty of individual APEX $NO_2$ VCDs ($\sigma_{\mathrm{VCD}} \approx 59\,\mu\mathrm{mol\,m^{-2}}$). In addition, airborne and ground-based instruments do not measure exactly the same air mass due to different viewing directions and different measurement times. Furthermore, the vertical

sensitivity to $NO_2$ is different for APEX and MAX-DOAS, so that incorrect assumptions about the vertical profile of $NO_2$ affects the comparison. It should be noted that APEX and MAX-DOAS observations are not fully independent, because the APEX $NO_2$ retrieval algorithm uses MAX-DOAS measurements as $\mathrm{VCD_{ref}}$.





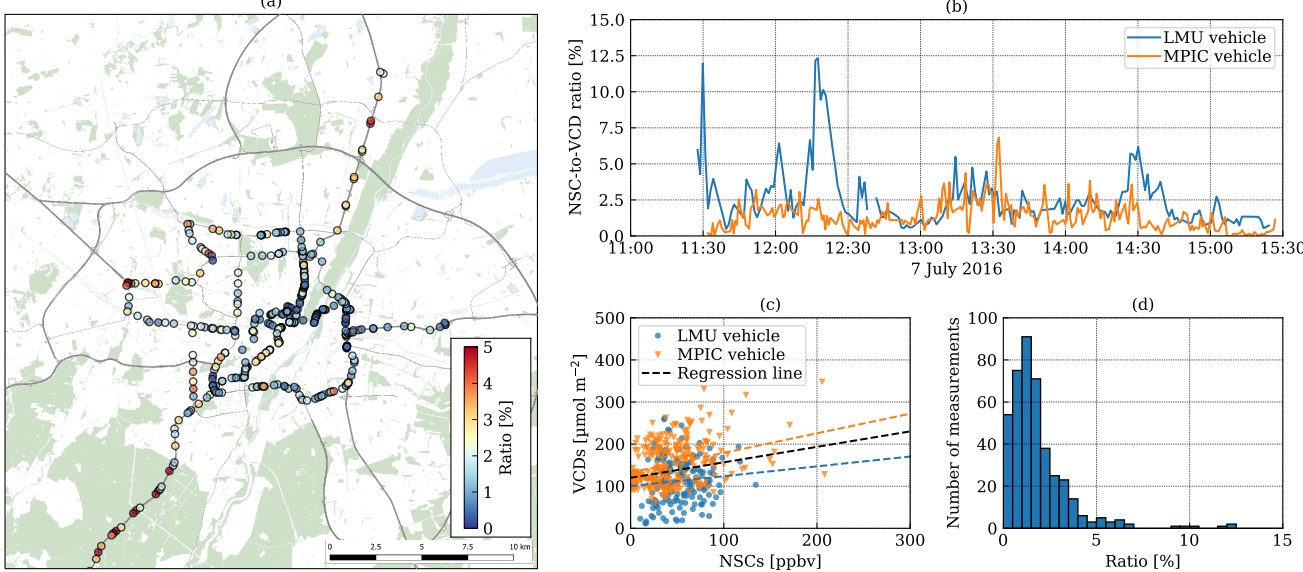

**Figure 8.** (a) Map of ratios of $NO_2$ near-surface concentrations (NSC in $\mu mol\,m^{-3}$) to vertical column densities (VCD in $\mu mol\,m^{-2}$) given in percent measured by in situ monitors and MAX-DOAS on-board the LMU and MPIC vehicle. (b) The time series of ratios measured with the two vehicles, (c) a scatter plot of NSCs and VCDs, and (d) a histogram of ratios. Map data from © OpenStreetMap contributors 2021. Distributed under the Open Data Commons Open Database License (ODbL) v1.0.

### 3.5 Comparison of near-surface concentrations and vertical column densities

The ratio of NSC and VCD links the $NO_2$ VCD retrieved from an imaging remote sensing instrument to the surface, which
is essential to access air pollution levels and as input for epidemiological studies. In the MuNIC campaign, both NSCs and VCDs were measured simultaneously by the in situ monitors and the MAX-DOAS instruments aboard the MPIC and LMU vehicle. We compute ratios per unit meter in percent from NSC (in $\mu mol\,m^{-3}$) and VCD (in $\mu mol\,m^{-2}$) measurements. Figure 8a shows the spatial distribution of ratios on 7 July 2016 (afternoon). Ratios are highest outside the city plume where VCDs are low but $NO_2$ NSCs on the roads are still high. In contrast, ratios are small inside the plume where both NSCs and VCDs
are high, but NSCs make up a smaller fraction of the total column. We did not measure NSCs away from roads, but since NSCs tend to be smaller away from roads and VCDs do not show much spatial variability, we expect lower ratios there. The scatter plot shows that VCDs increase slightly with NSCs but the scatter is very large (Fig. 8c) and the correlation is weak ($r = 0.22$). The histogram of all ratios has a median ratio of 1.4% with 90% of values ranging from 0.2% to 4.7%.

### 3.6 $NO_x$ and $CO_2$ emissions of point sources

The two largest point sources in the city are the Munich North CHP plant and Munich South CHP plant (see Fig. 1). For 2016, Munich North reported $CO_2$ and $NO_x$ mean emissions of $80\,kg\,CO_2\,s^{-1}$ and $53\,g\,NO_2\,s^{-1}$, respectively, and Munich South





reported $26 \, \mathrm{kg \, CO_2 \, s^{-1}}$ and $14 \, \mathrm{g \, NO_2 \, s^{-1}}$ (European Environment Agency, 2021). Since instantaneous emissions usually differ from annual means, we calculated the instantaneous emissions of Munich South using the natural gas consumption provided by the operating Startwerke München (SWM) using a conversion factor of $1.9225 \, \mathrm{kg \, CO_2}$ per $\mathrm{Nm^3}$ natural gas

(U.S. EPA, 2016) and the mass ratio of annual $CO_2$ and $NO_x$ emissions. The computed emissions are $44.2 \, \mathrm{kg \, CO_2 \, s^{-1}}$ and $23.1 \, \mathrm{g \, NO_2 \, s^{-1}}$, which is about 70% higher than annual means, and only vary slightly by $\pm 1.3 \, \mathrm{kg \, CO_2 \, s^{-1}}$ and $\pm 0.7 \, \mathrm{g \, NO_2 \, s^{-1}}$ between 9:30 and 14:00 UTC on 7 July 2016.

$CO_2$ and $NO_x$ emissions can be estimated from $CO_2$ and $NO_2$ observations. APEX flew once over Munich North at 12:27 UTC (Stripe 2) and twice over Munich South at 12:18 and 12:29 UTC (Stripe 1 and 2). The $NO_2$ emission plumes of the two

CHP plants are visible in the APEX observations (Fig. 9a,e and h). In addition, the $CO_2$ plume of Munich South was observed by the FTIR spectrometers (Fig. 10).

Figure 10a shows the setup of the two FTIR spectrometers where the instruments can observe $XCO_2$ upwind and downwind of the plume. The $XCO_2$ time series of the instruments is shown in Fig. 10c. The KIT instrument frequently observed the $CO_2$ enhancement of the CHP plant plume downwind of the source, while the TUM instrument observed the upwind values. Since

the plume location meanders depending on wind speed and direction, the line-of-sight of the FTIR instrument will not always include the plume. In fact, during the APEX measurements, the FTIR instruments did not capture any $CO_2$ enhancements, because the line-of-sight followed the sun towards the west, while the wind carried the plume towards the southeast. Figure 10b shows the APEX $NO_2$ field overlayed with the line-of-sights of the two FTIR spectrometers confirming that the plume was not in the instrument's field of view at APEX overpass time.

### 3.6.1    $NO_x$ emissions

The $NO_x$ emissions of Munich North and Munich South CHP plants were estimated from the APEX observations using a plume detection algorithm and mass-balance approach implemented as part of a Python package for "Data-driven Emission Quantification" (Kuhlmann, 2021a; Kuhlmann et al., 2021). The $NO_x$ emissions $Q$ were obtained by computing the cross-sectional flux as

$$Q = f(x) \cdot q(x) \cdot u(x) \cdot \cos(\alpha(x)) \tag{3}$$

where $x$ is the along-plume coordinate, $f$ is the $NO_2$:$NO_x$ conversion factor, $q$ is the line density and $u \cos(\alpha)$ is the wind speed normal to the cross section used for computing the line density. The line density was computed by fitting a Gaussian curve to the along-track $NO_2$ VCDs:

$$VCD(y) = \frac{q}{\sqrt{2\pi}\sigma} \exp\left(-\frac{(y-\mu)^2}{2\sigma^2}\right) + my + b \tag{4}$$

where $y$ is the across-plume coordinate, $\mu$ is a shift and $\sigma$ is the standard width. The $NO_2$ background was approximated by a linear function with slope $m$ and intercept $b$.

To compute the across- and along-plume coordinates, the location and extent of the $NO_2$ emission plumes was determined using a plume detection algorithm, which detects pixels where the local mean is significantly enhanced above the background





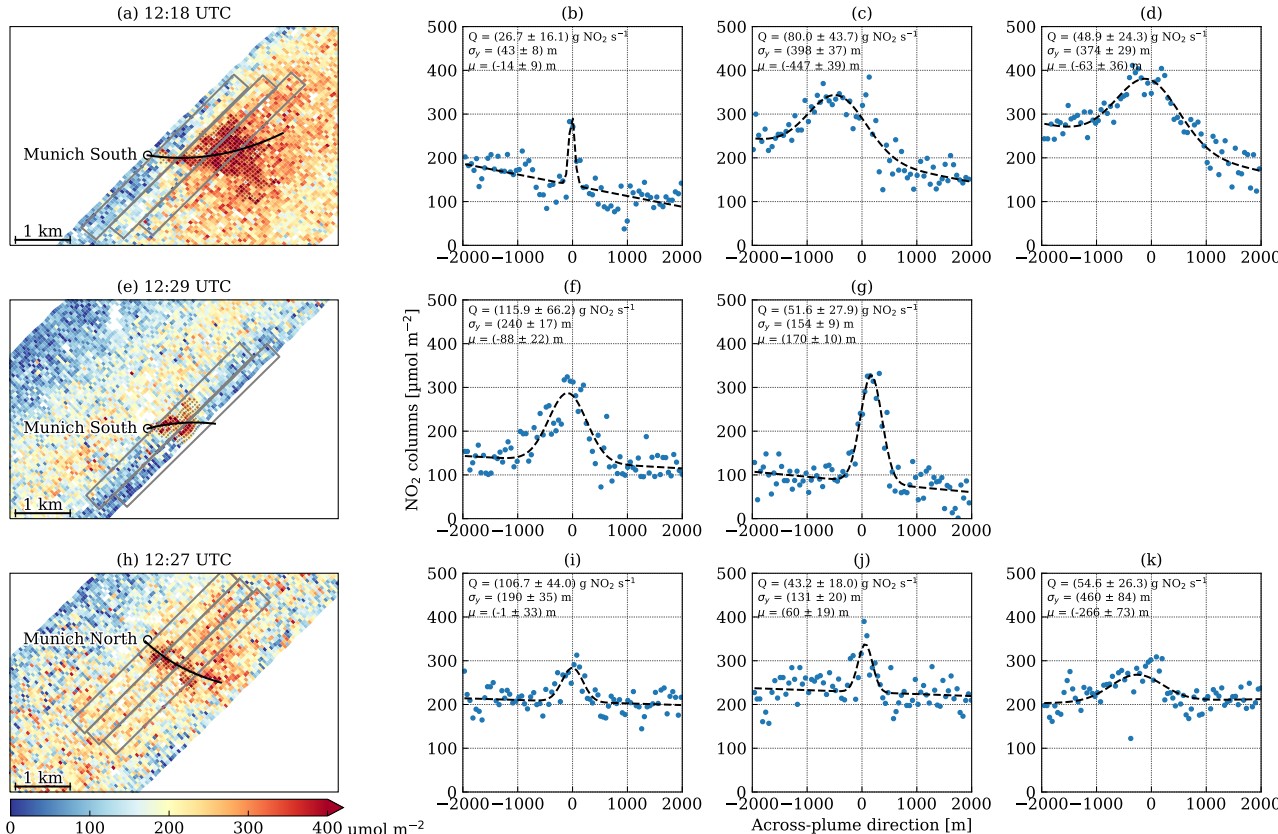

**Figure 9.** (a,e,h) APEX $NO_2$ maps with emission plumes (black dots) of the Munich South and Munich North CHP plant. The centerline of the plume is shown as black curve. (b-d,f,g,i-k) APEX $NO_2$ VCDs averaged in across-track direction within gray boxes shown in the maps and a Gaussian curve fitted to the observations (dashed line).

(Kuhlmann et al., 2019, 2021). The local mean was computed using a Gaussian filter with standard width of 0.75 pixels.

The background was computed as the median of 100 adjacent pixels in along-track direction to avoid issues with across-track stripes. A two-dimensional curve was then fitted to the detected pixels to describe the centerline of the plume. $x$ and $y$ coordinates were computed as arc length from the source and the distance from the curve, respectively. The tangent of the curve was used to determine the wind direction that is used to computed the angle $\alpha$.

A critical input for estimating the emissions is the wind speed profile, which determines the height of the plume due to

plume rise and the wind speed inside the plume. Figure 10e shows the time series of wind speeds and directions measured at the MIM's rooftop at 30 m above ground. The wind speed measured every minute shows high variability with a standard deviation of about $0.9 \, \mathrm{m \, s^{-1}}$ for 30-minute rolling means. Since MIM's rooftop is more than 3 km away from the stacks, the highly variable wind measurements are not well suited as input for estimating emissions. We therefore used simulations from the COSMO-1 and COSMO-7 analysis products provided by the Swiss Federal Office of Meteorology and Climatology



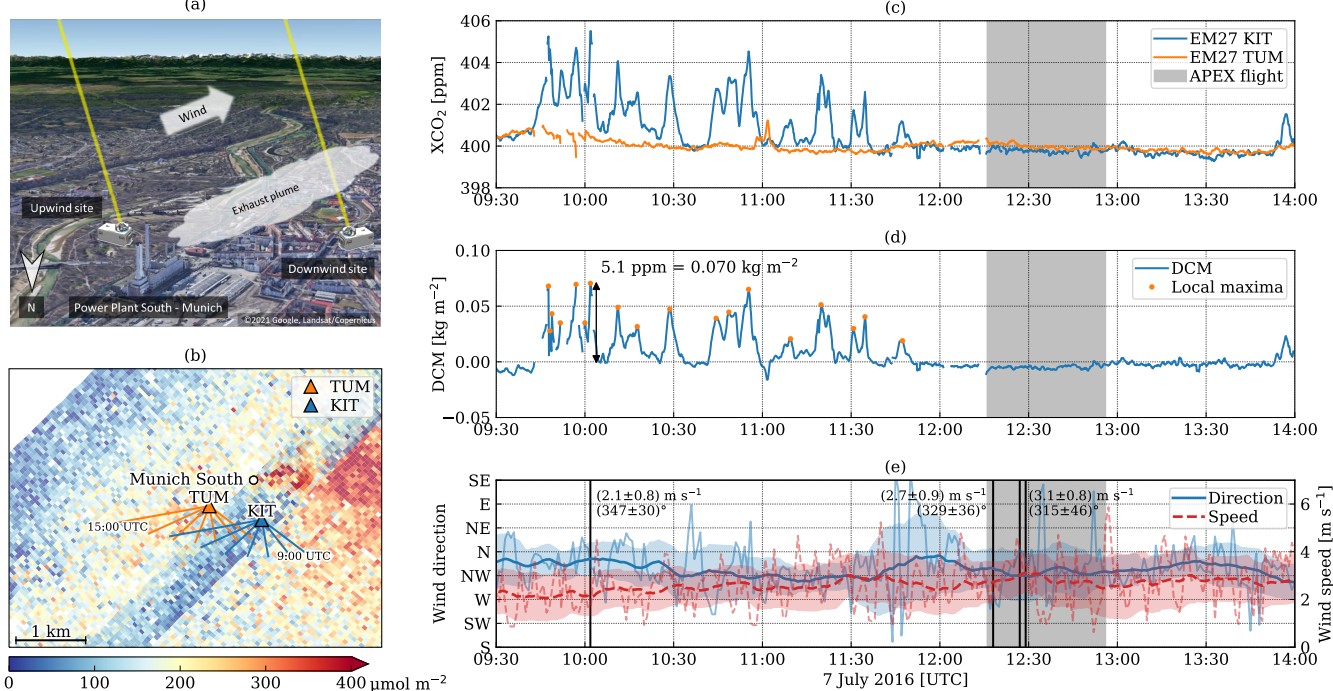

**Figure 10.** (a) The setup of the two FTIR spectrometers at Munich South. (b) APEX $NO_2$ map showing the $NO_2$ emission plume and the location of the FTIR spectrometers. The lines are hourly lines-of-sight below 1100 m above ground. (c) $XCO_2$ time series and (d) differential $CO_2$ measurements (DCM) for the two instruments. (e) Wind speed and direction measured every minute at LMU rooftop at 30 m above ground (thin lines). In addition, a 30-minute rolling average was applied to the time series (thick lines). The shaded area shows the $1\sigma$ temporal variability within 30 minutes. Vertical lines show times with the largest $CO_2$ peak and $NO_2$ measurements by the APEX instrument. The numbers are 30-minute mean wind speed and direction.

(MeteoSwiss) with 1 km and 7 km spatial resolution. The 10-m wind fields and vertical profiles are shown in Figs. S13 and S14. The COSMO-1 model with about 1 km resolution shows small convective cells with highly variable 10-m wind speeds over Munich with a similar spatial variability as the temporal variability of the ground measurements. Since the location of the convective cells is a stochastic processes, we average 6×6 grid cells to obtain an average wind profiles and its uncertainty at the stack locations.

The plume height needs to consider plume rise that can be computed from heat emissions, wind speed at the stack and the dispersion category (VDI - Fachbereich Umweltmeteorologie, 1985). Munich South CHP plant has three stacks ($H = 90$ m) for which we assume heat emissions of 70 MW. The dispersion category was already determined as very unstable (category: V) in Section 3.1. The wind speed at the height of the stacks is calculated from the COSMO-1 profiles as $2.5\pm1.3$ m s$^{-1}$ at 10 UTC. The plume height starts at the 90-m stack and raises quickly to a maximum height of 845 m at about 1580 m downstream of the stack (Fig. S15). We use the wind speed taken from the COSMO-1 profiles at the height of the plume to estimate $NO_x$





emissions. The distance from the plume is given by the arc length of the centerlines. We use the same heat emissions and stack height to estimate the plume rise for the Munich North CHP plant.

The $NO_2{:}NO_x$ conversion factor $f$ was computed using an NO lifetime of 0.3 hours for dispersion category V (VDI - Fachbereich Umweltmeteorologie, 2009, Section 10.2) and a residence time, which was computed using the distance from the 425 source to the cross section and the wind speed.

The uncertainties were computed using the uncertainty of the $NO_2$ VCDs in the Gaussian curve fit and the uncertainty of the wind speed from the spatial standard deviation in the COSMO-1 model. The uncertainty of the wind speed affects the uncertainty directly in Eq. (3) and indirectly due to its impact on the $NO_2{:}NO_x$ conversion factor.

Figure 9a, e and h show the three APEX observations at Munich North and Munich South with the detected pixels (black 430 dots) and centerlines. Line densities were computed by averaging five lines in across-track direction within ±2000 m from the centerline indicated by the rectangular boxes in the figure. The input parameters and the estimated $NO_x$ emissions are shown in Table S2 in the supplement. For Munich South, estimated $NO_x$ emissions range from 26.7 to 115.9 g $NO_2$ s$^{-1}$ with an average of 64.6±35.6 g $NO_2$ s$^{-1}$, i.e. a relative uncertainty of 55%. For Munich North, only three line densities could be computed resulting in emissions ranging from 43.2 to 106.7 g $NO_2$ s$^{-1}$ (average: 68.2±29.3 g $NO_2$ s$^{-1}$).

### 435 3.6.2 $CO_2$ emissions

The $CO_2$ emission strength can be estimated from the differential column measurements (DCM), i.e. the difference between the up- and downstream observations, which represents the enhancement due to the source (Fig. 10a, Chen et al. (2017)). If we assume that the emission plume can be approximated by a Gaussian plume model and that the local peaks in the DCM time series occur when the plume center moves over the downwind station (i.e. they correspond to the maxima in the Gaussian), the 440 source strength $Q$ would be given by

$$Q = \sqrt{2\pi}\, u(x)\, \sigma_y(x)\, DCM_{max}(x) \tag{5}$$

where $u$ is the wind speed inside the plume, $\sigma_y$ is the horizontal width of the plume and $DCM_{max}$ is the $CO_2$ enhancement in the center of the Gaussian plume. Note that $u$, $\sigma_y$ and $DCM_{max}$ depend on the distance from the source $x$.

Figure 10d shows the time series of DCM measured on 7 July 2016. In total, we identified 18 local maxima using a standard 445 peak finding algorithm of the SciPy library (Virtanen et al., 2020, Version 1.4.1) for estimating $CO_2$ emissions. The highest DCM was measured at 10:02 UTC with 5.1 ppm, which is converted to a mass column density of 0.070 kgm$^{-2}$.

The wind speed was taken from the COSMO-1 profiles to compute plume rise and the distance of the plume from the stack at the slant line-of-sight of the FTIR instrument (Fig. S15). The KIT EM27 spectrometer was located 582 m downstream of the stack. At 10:02 UTC, the line-of-sight of the instrument and the plume height intersect at a height of 615$^{+274}_{-177}$ m. Consequently, 450 the distance from the source $x$ is at 915$^{+485}_{-114}$ m downstream of the source. The dispersion coefficients are computed based on VDI guidelines and are assumed constant above 180 m. It is computed by an empirical equation $\sigma_y(x) = F \cdot x^f$ with $F = 0.671$ and $f = 0.903$ for the dispersion category V. Therefore, $\sigma_y(x)$ ranges was computed as 317$^{+85}_{-36}$ m at the intersection. The standard deviations are computed from the spatial variability in COSMO-1 wind speeds.





The $CO_2$ emissions are then computed by Eq. (5) using a wind speed in the plume center of 2.4±0.7 at 615 m above ground, which results in emissions of 134±49 kg $CO_2$ s$^{-1}$. The uncertainty was estimated using a DCM uncertainty of 0.1 ppm for 1-minute averages (Chen et al., 2016) as well as the uncertainties of $u(x)$ and $\sigma_y(x)$. The uncertainty budget is dominated by the uncertainty in the wind speed that accounts for 72% to the total uncertainty. The uncertainty of the dispersion coefficient $\sigma_y$ accounts for 27% of the total uncertainty, which also depends on the uncertainty of the wind speed, while the uncertainty of the DCM measurements in neglectable.

The emissions were computed for all 18 peaks and results are shown in Tab. S3 in the supplement. The estimated $CO_2$ emissions vary strongly ranging from 38 to 134 kg $CO_2$ s$^{-1}$. The mean and standard deviation of all 18 estimates is 81.2±29.8 kg $CO_2$ s$^{-1}$.

### 3.6.3 Comparison with reported emissions

$CO_2$ and $NO_x$ emissions of the Munich South CHP plant were estimated as 81.2±29.8 kg $CO_2$ s$^{-1}$ and 64.6±35.6 g $NO_2$ s$^{-1}$ significantly higher than the emissions of 44.2±1.3 kg $CO_2$ s$^{-1}$ and 23.1±0.7 g $NO_2$ s$^{-1}$ computed from fuel consumption. The $NO_x$-to-$CO_2$ emission ratios are similar between reported and estimated emissions.

Likely the main reason for the discrepancy between reported and estimated emissions is the unstable and highly turbulent atmospheric boundary layer resulting in low and highly variable wind speeds. Accurate knowledge of the wind speed is critical as it does not only affect the computation of the flux, but also the calculation of plume rise, dispersion coefficient and NO lifetime.

The method applied to estimating $CO_2$ emissions from the FTIR measurements assumed that the plume shape can be approximated by a Gaussian and that the local maxima are the maxima in the vertically integrated Gaussian plume. However, maxima in the time series also occur when the wind speed is below the average. As a result, the assumption of an average wind speed would overestimate the $CO_2$ emissions. The individual COSMO-1 wind profiles (Fig. S14) show that the lower wind speeds would halve the values, which would also lead to a halving of estimated $CO_2$ emissions. The turbulent flow will also result in puff-like structures in the plume where $CO_2$ values are locally enhanced or reduced compared to a Gaussian model, which will either over- or underestimate emissions. To limit the impact on our estimate, we have estimated emissions for all local peaks in the time series, yet our estimates are still likely to be too high, as the algorithm for identifying peaks does not account for minor peaks.

The computation of the cross-sectional flux for estimating $NO_x$ emissions from the APEX $NO_2$ observations is also limited by the turbulent flow, because the emission plumes are already very wide just 1 km downstream of the source, which makes it difficult to determine the centerline of the plume and the angle between APEX stripe and wind vector. The wind profiles from the simulations and the measurements are less consistent in the afternoon and the COSMO-1 wind profiles might be overestimated (Fig. S14), which could explain the overestimated $NO_x$ emissions. In addition, the estimation of $NO_x$ emissions is very sensitive to the $NO_2$-to-$NO_x$ conversion factor $f$, which in turn depends on NO lifetime and wind speed. For turbulent flow, the residence time is likely longer than the value computed from mean wind speed and distance from the source resulting in a smaller conversion factor and consequently lower estimated $NO_x$ emissions. Finally, it should also be noted that the VDI guidelines might not be sufficiently accurate for conditions on the campaign day.


## 4 Conclusions

In this paper, we presented results from the MuNIC measurement campaign conducted in July 2016 in Munich. The campaign
measured $NO_2$ near-surface concentrations (NSC) and vertical column densities (VCD) with stationary, mobile and airborne
in situ and remote sensing instruments. A central element of the campaign was a measurement flight with the APEX imaging
spectrometer mapping the spatial distribution of $NO_2$ VCDs on 7 July 2016 afternoon.

Our results confirm that airborne imaging remote sensing is suited for spatial mapping of $NO_2$ VCDs and the detection of
emission plumes from larger point sources. The obtained $NO_2$ VCDs retrievals agree well with mobile MAX-DOAS measure-
ments conducted on the same afternoon (r = 0.55) and estimated uncertainties are similar to previous findings (e.g. Tack et al.,
2017). The $NO_2$ map obtained from the APEX data could depict the coarse-scale $NO_2$ distribution with lower values upwind
and higher values downwind of the city but with no strong signatures of individual sources except for the two power plants.

We observed substantial differences and a weak correlation between VCDs and NSCs (r = 0.22). While APEX $NO_2$ VCDs
are not elevated along roads, NSCs shows high spatial and temporal variability along roads with highest values in congested
areas and tunnels. One reason for these differences is atmospheric mixing, but also 3D radiative transfer effects were recently
found to reduce the effective spatial resolution of the ground pixels and causing random uncertainties in the presence of
buildings as well as an underestimation of $NO_2$ VCDs due to building shadows (Schwaerzel et al., 2021).

The ratio of NSCs and VCDs is important when using imaging remote sensing for air pollution studies. However, the low
correlation between them demonstrates that more advanced methods (than a simple constant factor) need to be applied to
convert airborne $NO_2$ maps to near-surface concentrations. A way forward could be combining airborne observations with a
city-scale dispersion model that links NSCs and SCDs and also needs to consider 3D radiative transfer. The unique dataset
collected in the MuNIC campaign will be important for validating such approaches.

We found that $NO_x$ emission estimates of the two combined heat and power (CHP) plants are significantly higher than
reported values, while uncertainties are high due to high variability in wind speeds during the campaign day. For Munich
South, $NO_x$ emission estimates are consistent with $CO_2$ emissions determined from two ground-based FTIR instruments. The
accuracy of the estimates was limited by the low but highly variable wind speeds, which are required to compute the fluxes as
well as plume rise, dispersion and residence time that have a large impact on the estimated emissions.

Due to the difficulty of estimating emission from airborne imaging spectrometers under convective conditions, flying dur-
ing stable conditions, i.e. higher and less variable wind speeds and directions, should be an option to reduce uncertainties.
While conditions tend to be less convective in the early morning and late afternoon, such conditions increase other challenges
including large solar zenith angles, for which 3D radiative transfer effects can affect emission estimates (Schwaerzel et al.,
2020).

*Data availability.* The data used in this publication are available at the supplement.



*Author contributions.* The paper was written by GK with input from all co-authors. The APEX measurements were coordinated by AH. The

APEX Level-0, surface reflectance and auxiliary products were processed by AH. The APEX $NO_2$ product was retrieved and analysed by GK. The ASD measurements were conducted by AS, post-processed by AD, and analysed and compared to APEX by MS and GK. The CAPS measurement were analysed by GK. The CE-DOAS and LP-DOAS measurements were analysed by YZ, KLC and MW. The FTIR measurments were conducted, analysed and interpreted by JC, DHN and FD. The stationary MAX-DOAS measurements were analysed by KLC. The mobile MAX-DOAS measurements were conducted by SDo and SDö and analysed and interpreted by SDo, SDö and TW. The

MAX-DOAS instrument at OvM was provided by CL. The analysis and comparison of the data, the computation of the NSC and VCD ratios and the estimation of the emissions was conducted by GK. The MuNIC campaign was organized by GK, DB and MW. The study was conducted within the HighNOCS project supervised by BB.

*Competing interests.* Some authors are members of the editorial board of Atmospheric Measurement Techniques. The peer-review process was guided by an independent editor, and the authors have also no other competing interests to declare.

*Acknowledgements.* We would like to thank Sebastian Böhnke and Markus Garhammer for their support in driving the vehicles, Nan Hao and Zhouru Wang for assistance with setting up the MAX-DOAS instruments, Katharina Hauk for the support with the measurements and analysis of the mobile MAX-DOAS data. We acknowledge the Bavarian State Office for the Environment (LfU) for providing the $NO_2$ measurements from the monitoring network, the Stadtwerke München (SWM) for providing operation data for the Munich South CHP plant, and MeteoSwiss for providing the COSMO-1 and COSMO-7 analysis product. We are very thankful to Ludwig Heinle for setting up the

FTIR measurements, Frank Hase (KIT) for providing his EM27 instrument.

*Financial support.* The research has been supported by the Swiss National Science Foundation (SNSF) as part of the MuNIC and HighNOCS project (grant no. 170264 and 172533) as well as by Swiss University Conference and ETH board as part of the Swiss Earth Observatory Network (SEON).



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
