# Peer review of "Mapping the spatial distribution of NO2 with in situ and remote sensing instruments during the Munich NO2 imaging campaign"

_Atmospheric Measurement Techniques, 2021_

## Referee Comment (RC1)

**Review of: Mapping the spatial distribution of NO2 with in situ and remote sensing instruments during the Munich NO2 imaging campaign (Kuhlmann et al., 2021)**

The manuscript discusses results from the Munich NO2 imaging campaign (MuNIC) where nitrogen dioxide (NO2) near-surface concentrations (NSC) and vertical column densities (VCD) were measured with several stationary, mobile and airborne in situ and remote sensing instruments. The measurements have been intercompared with a focus on one day where also airborne APEX measurements were performed. Also a mass-balance approach is discussed to estimate NOx emissions of two CHP plants from the APEX observations. The scientific content of the paper fits well within the scope of AMT. The manuscript is well-written and generally well-structured. Therefore, I recommend its publication in AMT. However, a number of revisions (detailed below) need to be conducted in the paper before publication.

**General comments**

main comment: the different types of measurements are discussed in full detail. However, the study is very short on the actual comparison of airborne and in-situ observations and understanding of the relationship between VCDs and NSCs, while this is put forward as one of the main drivers of this study. It would also help the reader to more extensively explain why the relation NSC - VCD is not trivial and what are the main parameters affecting this relation, etc. As sect. 3.5 is not comparing with airborne data, mobile in-situ and remote sensing data could be better exploited by also taking into account measurements at other days than 7 July 2016, to see if findings are consistent or impacted by certain environmental parameters (I understood the campaign took place in the first two weeks of July). Also the LP-DOAS is described in this work but the data is not really exploited. This comment is linked to the following two comments.

p.3 I. 52: '...and to advance the understanding on the relationship between VCDs and NSCs in Munich, Germany'  $\rightarrow$  This is actually not well developed in this work. You could elaborate on the reasons why the relation is non-trivial. What is impacting it? etc.

p.3 l. 56: '...analysis the consistency of in situ and airborne NO2 observations'  $\rightarrow$  in-situ and airborne observations have not been compared in this work but mentioned here as an objective.

p.15 l.330: Key highways and key intersections/roundabouts could be identified in previous studies. In some cases a clear distinction could be made between the  $NO_2$  field upwind and downwind of the line source.

p.17 l.357: It is not fully clear how you compute the ratios: so basically it is giving the percentage of the VCD which is in the 1 m partial column at the surface? The percentage seems (very) low. Is this also a correct definition for the near-surface layer or  $NO_2$  NSC?

**Minor comments**

p.2 I.31: What you also might mention as an option is putting in-situ monitors on mobile platforms such as trams, buses, taxi cabs, etc. There are also some examples in the literature of putting in-situ monitors on Google street view cars, e.g. Apte JS, Messier KP, Gani S, Brauer M, Kirchstetter TW, Lunden MM, Marshall JD, Portier CJ, Vermeulen RCH, Hamburg SP. High resolution air pollution mapping with Google Street View cars: Exploiting big data. Environ. Sci. Tech, 2017.

p.5 l.77: The data is probably available per minute (for scientific use), but averaged per hour?

p.5 I.96: Please specify if a profiling is performed in each of the 7 azimuth directions or only in one main direction?

p.6 l.119: You might specify the major changes/updates of v2 compared to v1.

p.6 l.127: As I understand here, you used a different reference for each flight line. However, as the along-track drift was small for this flight you probably could use one reference for all flight lines?

p.7 l.148: if the spatial resolution is 80 m x 60 m, why do you upsample to 10 m?

p.8 l. 181: not fully clear: I assume integration time for 1 spectrum is shorter than 30 or 60 seconds, but you probably average an amount of spectra. Please clarify what is your integration time (for 1 spectrum) and total measurement time.

p.9 l.194: Please clarify here already that the instruments point towards the sun during measurements.

Section 3.1: Please reformulate 3.1 title to something more specific, something like 'campaign day with APEX overpass'

p.10 l.235: The absence of photolysis in the evening/night plays an important role as well.

p.10 l.248: The DoF should be taken into account to get an idea on how much layer information can be extracted

p.10 l.255: It would help the reader to clarify that the 2 sec. measurements are very sensitive to local effects, e.g. diesel truck next to the mobile in situ instrument.

Fig. 3: The stations of Landshuter Allee and Stachus were closely crossed by the mobile in situ. It would be nice to add the LfU station measurement to the time series (b) and/or (c) and the time of co-located measurement.

p.14 l. 320: Not clear to me why it is mentioned that uncertainties of the reference SCD is negligible. VCD errors from mobile measurements can be substantial (20-30%).

p.16 l.347: It is rather due to the relatively high uncertainty of both types of measurements, not only the airborne measurements.

**Technical corrections**

p.2 l.26: Because of these localized emissions and the relatively short lifetime of NO2,...

p.5 l.76: Fig. or Figure. Please make it consistent throughout the work.

p.9 l.206: R. in R. Richter can be removed?

p.9 l.212: an hand-held  $\rightarrow$  a hand-held

Figure 2 and other plots: indicate time is in UTC on the axis, in a similar way like done in S2 and S3.

p.23 l.495: ...and the detection of emission plumes from larger point sources and cities.

p.23 I.499: shows  $\rightarrow$  show

---

## Author Response (AR1)

**Mapping the spatial distribution of $NO_2$ with in situ and remote sensing instruments during the Munich $NO_2$ imaging campaign**

G. Kuhlmann et al. 2021

**Response to the Reviewer's Comments**

**Reviewer 1**

**General comments**

**Reviewer Point P 1.1** — This manuscript mainly analyzes data taken on the single day of 7 July 2016 during the MuNIC campaign to investigate the NO2 VCD spatial distribution field from an airborne imaging spectrometer (APEX) and evaluate its validity by comparisons with other observations such as mobile MAX-DOAS observations. Then, an attempt is made to estimate NOx emissions from combined heat and power plants. Since the accurate estimate of NOx emissions is highly demanded, the subject is appropriate for AMT. However, uncertainty in the critical parameters discussed (such as the APEX NO2 VCD values and NOx and CO2 emissions) is potentially too large to appropriately bring new findings and therefore conclusions. In addition, I identified some places that need much more clarification and discussions. After adequately addressing these and other concerns described below, I recommend that this manuscript will be published.

**Reply**: We like to thank reviewer #1 for their positive comments, critical assessment and useful points to improve the quality of our paper. In the following we address their concerns point by point showing changes in the paper in blue. We hope that we have clarified all concerns in the revised manuscript.

**Specific comments**

**Reviewer Point P 1.2** — I am concerned that uncertainty in the critical parameters discussed (such as the APEX NO2 VCD and NOx and CO2 emissions) is potentially too large to appropriately bring new findings and therefore conclusions. I think that the authors need more work to quantify those uncertainties more precisely by covering as much sources of uncertainty and, if possible, should attempt to reduce the total uncertainty, in order to draw conclusions through convincing discussion. The details will be given below.

**Reply**: We agree that the estimated emissions have high uncertainties and are likely overestimated. It is therefore not possible to improve the already quite accurate knowledge of emissions calculated from fuel consumption. We have revised the abstract, Section 3.6.3 and the conclusions for clarification. The main source of uncertainty is the large variability in the wind speed, which cannot be improved as more accurate wind information are not available for the campaign. We have added some additional discussion of smaller sources of uncertainty, but do not think it is necessary to add a detailed uncertainty budget for minor uncertainties. We have provided additional details in our replies to the specific points.

**Reviewer Point P 1.3** — I think that the term "VCD" used throughout this manuscript should be the tropospheric VCD. If so, please correct.

**Reply**: "VCD" refers to tropospheric vertical column densities. We have specified this in the manuscript.

**Reviewer Point P 1.4** — It is convenient for the readers to go through the manuscript without looking at supplement figures. Please consider to re-arrange all figures, including supplement figures.

**Reply**: We have moved some figures from the supplement to the manuscript (Fig. S3 and S7), but since the space in the main document is limited, some figures can only be in the supplement.

**Reviewer Point P 1.5** — In the abstract and conclusions, the authors state that "... agree well .., (r=0.55)". Also, it is stated in the text (L345-346) that "... agree quite well with a moderate correlation coefficient of 0.55." I think that the authors overstate there although a correlation coefficient was only 0.55. In addition to the correlation coefficient, the authors should discuss the difference to evaluate the agreement quantitatively.

**Reply**: We have changed the adjective for describing the correlation to "fair" and "moderate" in abstract, Section 3.4 and the conclusions. Furthermore, we have also computed mean bias and standard deviation of the differences between APEX and MAX-DOAS and modified the text accordingly:

In total, 518 co-located APEX and MAX-DOAS observations are available. APEX and MAX-DOAS $NO_2$ VCDs agree with a moderate correlation coefficient $r$ of 0.55. The regression line has a slope of 0.68 and an intercept of $48.3 \, \mu mol \, m^{-2}$. Furthermore, the mean bias (MB) computed from the differences between APEX and MAX-DOAS measurements is close to zero with $2.9 \, \mu mol \, m^{-2}$. The standard deviation (SD) of the differences is $58.1 \, \mu mol \, m^{-2}$. The discrepancy between airborne and ground-based observations is largely explainable by the relatively high uncertainty of individual APEX and MAX-DOAS $NO_2$ VCDs (i.e. APEX uncertainties were approximately $60 \, \mu mol \, m^{-2}$). [...]

**Reviewer Point P 1.6** — The last sentence of the abstract about the epidemiological studies should be removed since almost no relevant discussion has been made in the text.

**Reply**: We have removed the phrase "for example as input for epidemiological studies" from the abstract.

**Reviewer Point P 1.7** — In the abstract and the "2 Data and method" section, please provide information of latitude and longitude where the MuNIC campaign was conducted.

**Reply**: We have added information about longitude and latitude (48.1375°N, 11.575°E) to Section 2 (Data and method) and in some figures.

**Reviewer Point P 1.8** — L90-105: For the stationary MAX-DOAS, please state how the scaling factor of O4 has been treated in the aerosol extinction coefficient profile retrieval.

**Reply**: We added the following sentence to the manuscript:

Due to the systematic discrepancy between observation and model simulation of $O_4$ DSCDs (e.g., Wagner et al., 2009), all MAX-DOAS observations of $O_4$ DSCDs are multiplied with a correction factor of 0.8 (see Chan et al., 2019, for details).

**Reviewer Point P 1.9** — L103: What does the 1D-layer mean?

**Reply**: To clarify, we have reworded the sentence to "air mass factors for discrete vertical layers (1D-layer AMF)".

**Reviewer Point P 1.10** — L109: Please add a description how to estimate the along-track resolution of 6 m. I guess that the authors have considered the aircraft cruise speed and the integration time of spectrometer but the clarification would be helpful.

**Reply**: We added the following explanation:

The along-track resolution is computed from aircraft altitude, ground speed and integration time.

**Reviewer Point P 1.11** — L128: For the fitting window of 470-510 nm, H2O and O3 should be included in the DOAS analysis.

**Reply**: We have added the following explanation and references explaining why $H_2O$ and $O_3$ are not included:

The cross sections of $O_3$ and $H_2O$ are not considered due to cross correlations and overparameterization in the small fitting window (Popp et al., 2012; Tack et al., 2017).

**Reviewer Point P 1.12** — L132, Eq 1: Please add an explanation for readers to readily understand this equation.

75   **Reply**: We have added a second equation and some more details to make equation better understandable.

**Reviewer Point P 1.13** — L171-174: Please discuss the difference between CAPS and CE-DOAS data, in addition to the correlation coefficient.

**Reply**: We also now compute the root mean square errors (RMSE), updated Fig. S2 and modified the text as:

> CAPS and CE-DOAS agreed well when operated together on the LMU vehicle on 1 and 4 July 2016.
80   Pearson correlation coefficients were 0.995 and 0.984 and root mean square errors (RMSE) were 6.1 and 5.9 ppbv on the two days. The difference between CAPS and CE-DOAS are due to instrument differences such as length of tubing and integration times, the short measurement interval of 2 seconds, and the high variability of $NO_2$ concentrations on roads. Furthermore, it was necessary to shift CAPS measurements by 16.9 s and 5.6 s, respectively (Fig. S2). The time differences were caused by non-synchronised computer
85   clocks and the differences in tubing and integration times. On campaign day, GPS times were used to minimize time differences between instruments.

**Reviewer Point P 1.14** — L211: Please elaborate what the surface hemispheric-conical reflectance factor HCRF.

**Reply**: We have slightly modified the paragraph and added a reference providing more details on HCRFs:

> The APEX reflectance product calculates hemispheric-conical reflectance factors (HCRF; Schaepman-
90   Strub et al., 2006) using the atmospheric correction software ATCOR-4 (Richter and Schläpfer, 2002).

**Reviewer Point P 1.15** — L230: Please add a description about the physical processes behind this dispersion category, which was estimated as very unstable from wind speed, cloud fraction, and hour of day.

**Reply**: The VDI guidelines determine dispersion categories based on tabulated values for wind speed, cloud cover and
95   hour of day. We have modified the sentence as follows:

> We estimated the dispersion category during daytime as very unstable (category V) based on the procedure published by the Association of German Engineers, which determines dispersion categories based on tabulated values for wind speed, cloud fraction and hour of day (VDI - Fachbereich Umwelt-meteorologie, 2009, Annex A).

100   **Reviewer Point P 1.16** — L233-234: I do not think that Figure 2a shows a clear diurnal cycle with a morning peak and an increase in the evening.

**Reply**: We deleted the phrase "a clear diurnal cycle with".

**Reviewer Point P 1.17** — L255-258: I could not understand what authors want to say in this paragraph.

**Reply**: We have revised the paragraph as follow for clarification:

105
      $NO_2$ mole fractions varied strongly along the route, because they are very sensitive to local emissions. The values ranged from 0 to 890 ppb with highest values being observed in congested areas, in front of traffic light or in tunnels. While 2-second values show high variability along the road, 30-minute averages are similar to hourly measurements at the roadside stations at Stachus and Landshuter Allee when the vehicle passed at distances of 75 m and <10 m from the stations, respectively (Fig. 4b and c). A more
110 detailed comparison of mobile measurements and measuring stations for these data can be found in Zhu et al. (2020).

**Reviewer Point P 1.18** — L267: How much FWHM values were indicated by in-flight and laboratory calibrations? Please add these info in the text.

**Reply**: APEX FWHMs obtained during laboratory calibration are provided in Sec. 2.1:

115
      To retrieve $NO_2$ VCDs, spectra were acquired in the unbinned mode providing the highest nominal spectral resolution of 0.86 nm full-width-at-half-maximum (FWHM) and 0.45 nm spectral sampling interval (SSI) in the VNIR channel with 334 spectral bands. Note that FWHM and SSI vary with wavelength and range from 0.86 to 15 nm (FWHM) and 0.45 to 7.5 nm (SSI), owing to the dispersion characteristics of the VNIR prism.

120 FWHMs obtained from in-flight calibration are about 30% higher than lab calibration. We added a reference to Sec. 2.5 in the text.

**Reviewer Point P 1.19** — L354-355: It was hard to understand why it is essential as input for epidemiological studies.

**Reply**: Epidemiological studies, which access the impact of NO2 concentrations on human health, require maps of NO2
125 NSCs. We have revised the section.

**Reviewer Point P 1.20** — L374-375 and Figures 9a, e, and h: To confirm that the high APEX NO2 areas were plumes from CHP plants, please add figures of the wind field and its discussion here.

**Reply**: We have added stream lines to the figures showing COSMO-1 and COSMO-7 wind fields. They show that the mean wind direction is consistent with the direction of the plumes (Fig. S11). The wind direction in the model fields is

130 also consistent with the measurements at the MIM's rooftop and at the OvM tower. We are now introducing the COSMO wind fields in Section 3.1 before discussing the plumes in the APEX data.

**Reviewer Point P 1.21** — L403-404: How accurate is the determined wind direction? How is the consistency with the wind field?

**Reply**: The measured and simulated wind directions are consistent but highly variable with a standard deviation of 46°
135 for a 30-minute period as can be seen in Fig. 12e.

**Reviewer Point P 1.22** — L417: I cannot evaluate how reasonable the assumed heat emissions of 70MW are.

**Reply**: We added the missing information that values for heat emissions were provided by the CHP plant operator Stadtwerke München.

**Reviewer Point P 1.23** — L426-428: Potentially critical sources of uncertainties, such as heat emission, NO2:NO
140 conversion factor, and residence time, have been missing in the uncertainty estimate here.

**Reply**: An uncertainty of 10% in heat emissions results in an uncertainty in plume height and plume distance of $11\,\mathrm{m}$ and $19\,\mathrm{m}$, respecitvely, which has not been included as it is an order of magnitude smaller than the uncertainty due to the wind speed (Fig. S13). The $NO_2:NO_x$ conversion factor and its uncertainty is computed from the residence time and its uncertainty is also strongly impacted and very likely dominated by the wind speed uncertainty. The values and their
145 uncertainties are available in Table S2.

**Reviewer Point P 1.24** — L433: "35.6 g NO2 s-1" or "55%" is not an uncertainty but just the variation of the average. Please provide the uncertainty estimated considering as much sources of uncertainties as possible.

**Reply**: The value of $35.6\,\mathrm{g\,NO_2\,s^{-1}}$ is the mean of the uncertainties, which were given in Table S2. The uncertainties were calculated considering uncertainties in APEX $NO_2$ VCDs, wind speed and direction, and $NO_2:NO_x$ ratio. The standard
150 deviation is very similar with $34.4\mathrm{g\,NO_2\,s^{-1}}$ showing that our uncertainty estimate should include the most important sources of uncertainty.

Since it would actually be more appropriate to state the standard error of the mean instead of the mean uncertainty, we have revised the sentence as:

$CO_2$ and $NO_x$ emissions of the Munich South CHP plant were estimated as $81.2\pm8.6\,\mathrm{kg\,CO_2\,s^{-1}}$ and
155 $64.6\pm15.9\,\mathrm{g\,NO_2\,s^{-1}}$ significantly higher than...

**Reviewer Point P 1.25** — L464: I wondered that the estimated emissions were higher than those computed from fuel consumption because this work picked up peak values only. Moreover, I am afraid that the estimated emission depends on how peak values picked up.

**Reply**:  Yes, the method depends on how peak values are picked, because the method requires that peak values are the maximum in the Gaussian plume model, which has some limitations as already discussed and slightly revised in Section 3.6.3:

> The method applied to estimating $CO_2$ emissions from the FTIR measurements assumed that the plume can be approximated by a Gaussian plume model and that the local maxima are the maxima in the vertically integrated Gaussian plume. However, maxima in the time series also occur when the wind speed is below the average. As a result, the assumption of an average wind speed would overestimate the $CO_2$ emissions. The individual COSMO-1 wind profiles (Fig. S12) show that the lower wind speeds would halve the values, which would also lead to a halving of estimated $CO_2$ emissions much better consistent with our estimate from fuel consumption. The turbulent flow will also result in puff-like structures in the plume where $CO_2$ values are locally enhanced or reduced compared to a Gaussian model, which will either over- or underestimate emissions. Furthermore, it is possible that only the edge of plume passes briefly through the line-of-sight, which would result in a local maximum but underestimate the emissions. To limit the impact on our estimate, we have estimated emissions for all local peaks in the time series, yet our estimates are still likely to be too high, as the algorithm for identifying peaks does not account for minor peaks.

**Reviewer Point P 1.26**  —  L466-469: I could not evaluate the sentences. This is because I was skeptical if the discrepancy the authors state was really a meaningful difference while 1) the estimated emissions has potentially a large uncertainty (as discussed at L470-487) and 2) conditions for estimates of reported and estimated emissions were unlikely identical. Moreover, I did not see any reason why the main reason is the unstable and highly turbulent atmospheric boundary layer, while there are other significant error sources (as discussed at L470-487).

**Reply**:  We have revised the sentence considering the comment (see previous reply).

**Reviewer Point P 1.27**  —  L476-477: The authors should estimate the emissions by incorporating minor peaks and add its quantitative discussion. I guess that the authors can identify minor peaks (at least some of them) manually without the algorithm.

**Reply**:  The algorithm already identifies 18 minor and major peaks (see Fig. 12d). If more minor peaks are included, we might pick peaks where only the plume edge moves through the line-of-sight, which would underestimate the emissions. We added this information to Section 3.6.3.

**Reviewer Point P 1.28**  —  L483-486: If the estimation of NOx emission is sensitive, the authors must include sensitivity analyses to the NO2-to-NOx conversion factor and residence time and their quantitative discussion in the manuscript.

**Reply**: We have revised the uncertainty calculation and include now a rough estimate of the uncertainty of NO lifetime of 50%, which roughly doubles the uncertainty of the conversion factor. A better estimate of the NO lifetime and its uncertainty would require $NO_x$ chemistry simulations for the turbulent emission plume, which is outside the scope of this study.

**Reviewer Point P 1.29** — L498: To support this argument, please discuss by referring to Irie et al. (2011) , who showed the relationship between the partial VCD (at 0-1 km) and NSCs.

**Reply**: We have revised the analysis of the relationship between NSCs and VCDs, added some additional information and cite Irie et al. (2011) now.

**Technical corrections**

**Reviewer Point P 1.30** — L1: "NO2" should be defined first. So, "... the Munich NO2 imaging ..." should be "... the Munich nitrogen dioxide (NO2) imaging ..."

**Reply**: Done.

**Reviewer Point P 1.31** — L49-50: I had an impression that only the issue that typical urban monitoring network is too small has posed the reliable retrieval of NSCs with airborne remote sensing observations. There should be other issues. Please discuss them also here.

**Reply**: We have changed the sentence to make it clear that a single measurement flight also limits the amount of data for training:

> These algorithms require large training datasets and cannot be applied to airborne measurement campaigns as typical urban monitoring networks are too small to train the relationship between VCDs and NSCs from a single measurement flight.

**Reviewer Point P 1.32** — Fig. 1: Please provide information of latitude, longitude, and direction.

**Reply**: We have added grid lines showing longitude and latitude to the figure.

**Reviewer Point P 1.33** — L162: Please write what API is.

**Reply**: "Teledyne API" is the registered name of a company, so it should not be changed.

**Reviewer Point P 1.34** — L191: Please write what KIT is.

215 **Reply**: Changed to "Karlsruhe Institute of Technology (KIT)".

**Reviewer Point P 1.35** — L192: Please write what TUM is.

**Reply**: Changed to "Technical University of Munich (TUM)".

**Reviewer Point P 1.36** — L206: "R.Richter" should be "Richter"

**Reply**: Done.

220 **Reviewer Point P 1.37** — L219: Should "... Munich based ..." be "... Munich is based ..."?

**Reply**: Done.

**Reviewer Point P 1.38** — Fig 2: Please re-write the figure caption for readers to readily understand that Fig. 2a is for monitoring stations and Fig. 2b is for LP-DOAS.

**Reply**: Done.

225 **Reviewer Point P 1.39** — Figure 4: Please provide information of latitude, longitude, and direction. Also, please indicate where CHP plants are located.

**Reply**: The data are not geo-referenced and thus showing latitude and longitude is not possible. However, we added "see Fig. 1" to the caption, which shows the location of the second APEX stripe. We also added a RGB image for orientation.

**Reviewer Point P 1.40** — Figure 6a: Please provide information of latitude, longitude, and direction.

230 **Reply**: see previous reply

**Reviewer Point P 1.41** — L350: Better to write "inconsistent" instead of "incorrect"?

**Reply**: Done.

**Reviewer 2**

**Reviewer Point P 2.1** — The manuscript discusses results from the Munich NO2 imaging campaign (MuNIC)
235 where nitrogen dioxide (NO2) near-surface concentrations (NSC) and vertical column densities (VCD) were measured with several stationary, mobile and airborne in situ and remote sensing instruments. The measurements have been

intercompared with a focus on one day where also airborne APEX measurements were performed. Also a mass-balance approach is discussed to estimate NOx emissions of two CHP plants from the APEX observations. The scientific content of the paper fits well within the scope of AMT. The manuscript is well-written and generally well-structured. Therefore, I recommend its publication in AMT. However, a number of revisions (detailed below) need to be conducted in the paper before publication.

**Reply**: We thank Frederic Tack for his positive comments and critical assessment, which improved the quality of the manuscript. In the following we address his concerns point by point showing changes in the paper in blue. We hope that we have clarified all concerns in the revised paper.

**General comments**

**Reviewer Point P 2.2** — main comment: the different types of measurements are discussed in full detail. However, the study is very short on the actual comparison of airborne and in-situ observations and understanding of the relationship between VCDs and NSCs, while this is put forward as one of the main drivers of this study.

**Reply**: Thank you for the comment. We have revised the objective to clarify that the main objective of this paper is the validation of the APEX observations and collection of data:

> We therefore conducted the Munich $NO_2$ Imaging Campaign (MuNIC) to validate airborne imaging spectrometers with ground-based observations and to collect data for advancing the understanding on the relationship between VCDs and NSCs in Munich, Germany. [...] In this paper, we validate the APEX $NO_2$ map using the mobile MAX-DOAS observations, analysis the consistency and relationship between $NO_2$ VCDs and NSCs, and demonstrate the applicability of the collected data to estimate the emissions of the two largest point sources in the city.

**Reviewer Point P 2.3** — It would also help the reader to more extensively explain why the relation NSC - VCD is not trivial and what are the main parameters affecting this relation, etc.

**Reply**: We have revised the section and added the following details:

> Airborne $NO_2$ maps from imaging remote sensing instruments can be valuable for studying the spatial distribution of air pollutants in a city. However, to access the impact of $NO_2$ on human health, it is necessary to transfer VCDs to NSCs using a transfer parameter $t$ such that:
>
> $$NSC = t \cdot VCD. \tag{1}$$
>
> The parameter $t$ (in $m^{-1}$) is the ratio of NSC (in $\mu mol\,m^{-3}$) and VCD (in $\mu mol\,m^{-2}$). It depends on the shape of the vertical $NO_2$ profile, which varies in space and time with local $NO_x$ emissions, meteorological

factors such as wind speed and vertical mixing, $NO_x$ chemistry and $NO_2$ advected from the surroundings.

**Reviewer Point P 2.4** — As sect. 3.5 is not comparing with airborne data, mobile in-situ and remote sensing data could be better exploited by also taking into account measurements at other days than 7 July 2016, to see if findings are consistent or impacted by certain environmental parameters (I understood the campaign took place in the first two weeks of July). Also the LP-DOAS is described in this work but the data is not really exploited. This comment is linked to the following two comments.

**Reply**: Yes, additional observations on other days are also available. Although it is certainly valuable to use these observations for a detailed evaluation of the NSC and VCD, we consider such an analysis to be outside the scope of this paper. We also note that the work of Zhu et al. (2020) already analyses the NSCs from a longer time series (including this campaign) and also uses the LP-DOAS measurements.

**Reviewer Point P 2.5** — p.3 l. 52: '...and to advance the understanding on the relationship between VCDs and NSCs in Munich, Germany' -> This is actually not well developed in this work. You could elaborate on the reasons why the relation is non-trivial. What is impacting it? etc.

**Reply**: see our previous reply

**Reviewer Point P 2.6** — p.3 l. 56: '...analysis the consistency of in situ and airborne NO2 observations' -> in-situ and airborne observations have not been compared in this work but mentioned here as an objective.

**Reply**: see our previous reply

**Reviewer Point P 2.7** — p.15 l.330: Key highways and key intersections/roundabouts could be identified in previous studies. In some cases a clear distinction could be made between the NO2 field upwind and downwind of the line source.

**Reply**: We have revised the sentence as follows:

While local enhancements related to traffic such as key intersections and highways could be identified in previous APEX campaigns (Popp et al., 2012; Tack et al., 2017, e.g.,), such structures are missing in Munich likely due to the high variability in wind speed and direction resulting in strong spatial mixing.

**Reviewer Point P 2.8** — p.17 l.357: It is not fully clear how you compute the ratios: so basically it is giving the percentage of the VCD which is in the 1 m partial column at the surface? The percentage seems (very) low. Is this also a correct definition for the near-surface layer or NO2 NSC?

**Reply**:  We have revised Section 3.5 improving the explanation of the computation of the ratios. Yes, the ratio if the

295   percentage of the partial NO$_2$ column in a 1-m layer near the surface. Having about 1-2% of the tropospheric NO$_2$ column in a only 1-m thick layer near the surface, seems to us a reasonable value.

   We also have slightly changed the methodology and now only fit a constant transfer parameter $t$ instead of a linear function. To discuss the result, we also use RMSE now, which is more meaningful than the correlation coefficient.

**Minor comments**

300   **Reviewer Point P 2.9** — p.2 l.31: What you also might mention as an option is putting in-situ monitors on mobile platforms such as trams, buses, taxi cabs, etc. There are also some examples in the literature of putting in-situ monitors on Google street view cars, e.g. Apte JS, Messier KP, Gani S, Brauer M, Kirchstetter TW, Lunden MM, Marshall JD, Portier CJ, Vermeulen RCH, Hamburg SP. High resolution air pollution mapping with Google Street View cars: Exploiting big data. Environ. Sci. Tech, 2017.

305   **Reply**:  We added using public buses or trams as further example in the introduction:

   although mobile sensors on public buses or trams could increase the measurement density (e.g., Hagemann et al., 2014; Hundt et al., 2018).

**Reviewer Point P 2.10** — p.5 l.77: The data is probably available per minute (for scientific use), but averaged per hour?

310   **Reply**:  This is possible. We have not requested data with higher temporal resolution, as this was not necessary for this study.

**Reviewer Point P 2.11** — p.5 l.96: Please specify if a profiling is performed in each of the 7 azimuth directions or only in one main direction?

**Reply**:  In this study, only measurements at an azimuth angle of 0°(pointing northwards) were analysed in the MAX-DOAS

315   retrieval.

**Reviewer Point P 2.12** — p.6 l.119: You might specify the major changes/updates of v2 compared to v1.

**Reply**:  We have revised the paragraph adding the following information:

   APEX NO$_2$ retrieval algorithms were developed at Empa in Switzerland (Popp et al., 2012) and at BIRA in Belgium (Tack et al., 2017) using the QDOAS software for the DOAS analysis and the LIDORT

320   radiative transfer model for AMF computations. In this study, we used a new version of the Empa APEX

NO$_2$ retrieval algorithm, which has been completely rewritten using Python scripts allowing for automatic and parallel processing of APEX measurements. The new version uses the flexDOAS Python library for the DOAS analysis (Kuhlmann, 2021) and the Monte Carlo MYSTIC solver for AMFs (Schwaerzel et al., 2020).

**Reviewer Point P 2.13** — p.6 l.127: As I understand here, you used a different reference for each flight line. However, as the along-track drift was small for this flight you probably could use one reference for all flight lines?

**Reply**: Since the spectral calibration also varies in across-track direction, due to the spectral smile of the instrument, it is not recommendable to use the same reference for all flight lines, because this would require to interpolate the spectra to the reference (or vice versa), which results in undersampling errors as APEX has a sampling ratio close to one.

**Reviewer Point P 2.14** — p.7 l.148: if the spatial resolution is 80 m x 60 m, why do you upsample to 10 m?

**Reply**: The APEX data are mapped on a longitude-latitude grid for comparing with the other measurements, for visualization and for applying the bias correction between stripes.

**Reviewer Point P 2.15** — p.8 l. 181: not fully clear: I assume integration time for 1 spectrum is shorter than 30 or 60 seconds, but you probably average an amount of spectra. Please clarify what is your integration time (for 1 spectrum) and total measurement time.

**Reply**: Thanks for pointing out this misunderstanding. Because of the different signal-to-noise ratios, the total measurements times were set to 30 s for the Tube MAX-DOAS and 60 s for the Mini-MAX-DOAS instruments, respectively. The obtained spectra are averages of multiple single measurements. The single integration times for these individual spectra are adjusted according to the current sky conditions for each (total) measurement to have a constant saturation level for each individual spectrum. Finally, a certain amount of individual spectra is averaged to obtain the prescribed total measurement times. For both instruments single integration times were mainly below 100 ms on the presented measurement day which was a rather clear day. Mean individual integration times were roughly 65 ms leading to number of scans of roughly 460 and 920 for the two instruments, respectively.

**Reviewer Point P 2.16** — p.9 l.194: Please clarify here already that the instruments point towards the sun during measurements.

**Reply**: Done.

**Reviewer Point P 2.17** — Section 3.1: Please reformulate 3.1 title to something more specific, something like 'campaign day with APEX overpass'

**Reply**: Done.

**Reviewer Point P 2.18** — p.10 l.235: The absence of photolysis in the evening/night plays an important role as well.

**Reply**: We added "...solar radiation is missing...".

**Reviewer Point P 2.19** — p.10 l.248: The DoF should be taken into account to get an idea on how much layer information can be extracted

**Reply**: We added the following sentences to the manuscript:

> The averaging kernels of MAX-DOAS retrieval for the lowest layer range between 0.85 and 1, indicating the retrieval reconstruct the lowest layer quite well. However, the lowest layer of the MAX-DOAS represents the average of the lowest 200 m, while LP-DOAS measures at about 30 m above ground. Therefore, it is expected the LP-DOAS measures higher concentration than the MAX-DOAS.

**Reviewer Point P 2.20** — p.10 l.255: It would help the reader to clarify that the 2 sec. measurements are very sensitive to local effects, e.g. diesel truck next to the mobile in situ instrument.

**Reply**: We have added the information that NO2 concentrations are "very sensitive to local emissions."

**Reviewer Point P 2.21** — Fig. 3: The stations of Landshuter Allee and Stachus were closely crossed by the mobile in situ. It would be nice to add the LfU station measurement to the time series (b) and/or (c) and the time of co-located measurement.

**Reply**: We have added markers showing the measurements at the stations when mobile measurements pass the stations. Note that Zhu et al. (2020) compare mobile and network observations in more details.

**Reviewer Point P 2.22** — p.14 l. 320: Not clear to me why it is mentioned that uncertainties of the reference SCD is negligible. VCD errors from mobile measurements can be substantial (20-30%).

**Reply**: We changed this to "is small compared to the other two components."

**Reviewer Point P 2.23** — p.16 l.347: It is rather due to the relatively high uncertainty of both types of measurements, not only the airborne measurements.

**Reply**: We changed this to

> The discrepancy between airborne and ground-based observations is largely explainable by the relatively high uncertainty of individual APEX and MAX-DOAS $NO_2$ VCDs (e.g. APEX uncertainties were approximately $60\,\mathrm{\mu mol\,m^{-2}}$). In addition, ...

**Technical corrections**

**Reviewer Point P 2.24** — p.2 l.26: Because of these localized emissions and the relatively short lifetime of NO2,...

**Reply**: Done.

380 **Reviewer Point P 2.25** — p.5 l.76: Fig. or Figure. Please make it consistent throughout the work.

**Reply**: We checked all uses of "Fig." and "Figure" and usage should be consistent with journal guidelines using "Fig." except at the beginning of a sentence.

**Reviewer Point P 2.26** — p.9 l.206: R. in R. Richter can be removed?

**Reply**: Done.

385 **Reviewer Point P 2.27** — p.9 l.212: an hand-held -> a hand-held

**Reply**: Done.

**Reviewer Point P 2.28** — Figure 2 and other plots: indicate time is in UTC on the axis, in a similar way like done in S2 and S3.

**Reply**: Done.

390 **Reviewer Point P 2.29** — p.23 l.495: ...and the detection of emission plumes from larger point sources and cities.

**Reply**: Done.

**Reviewer Point P 2.30** — p.23 l.499: shows -> show

**Reply**: Done.

**References**

Chan, K. L., Wang, Z., Ding, A., Heue, K.-P., Shen, Y., Wang, J., Zhang, F., Shi, Y., Hao, N., and Wenig, M.: MAX-DOAS measurements of tropospheric $NO_2$ and HCHO in Nanjing and a comparison to ozone monitoring instrument observations, Atmospheric Chemistry and Physics, 19, 10051–10071, https://doi.org/10.5194/acp-19-10051-2019, 2019.

Hagemann, R., Corsmeier, U., Kottmeier, C., Rinke, R., Wieser, A., and Vogel, B.: Spatial variability of particle number concentrations and $NO_x$ in the Karlsruhe (Germany) area obtained with the mobile laboratory 'AERO-TRAM', Atmospheric Environment, 94, 341–352, https://doi.org/https://doi.org/10.1016/j.atmosenv.2014.05.051, 2014.

Hundt, P. M., Müller, M., Mangold, M., Tuzson, B., Scheidegger, P., Looser, H., Hüglin, C., and Emmenegger, L.: Mid-IR spectrometer for mobile, real-time urban $NO_2$ measurements, Atmospheric Measurement Techniques, 11, 2669–2681, https://doi.org/10.5194/amt-11-2669-2018, https://amt.copernicus.org/articles/11/2669/2018/, 2018.

Kuhlmann, G.: flexDOAS – A flexible Python library for DOAS analysis, https://gitlab.com/empa503/remote-sensing/flexdoas, last access: January 7, 2022, 2021.

Popp, C., Brunner, D., Damm, A., Van Roozendael, M., Fayt, C., and Buchmann, B.: High-resolution NO2 remote sensing from the Airborne Prism EXperiment (APEX) imaging spectrometer, Atmospheric Measurement Techniques, 5, 2211–2225, https://doi.org/10.5194/amt-5-2211-2012, http://www.atmos-meas-tech.net/5/2211/2012/, 2012.

Richter, R. and Schläpfer, D.: Geo-atmospheric processing of airborne imaging spectrometry data. Part 2: Atmospheric/topographic correction, International Journal of Remote Sensing, 23, 2631–2649, https://doi.org/10.1080/01431160110115834, 2002.

Schaepman-Strub, G., Schaepman, M., Painter, T., Dangel, S., and Martonchik, J.: Reflectance quantities in optical remote sensing—definitions and case studies, Remote Sensing of Environment, 103, 27–42, https://doi.org/https://doi.org/10.1016/j.rse.2006.03.002, 2006.

Schwaerzel, M., Emde, C., Brunner, D., Morales, R., Wagner, T., Berne, A., Buchmann, B., and Kuhlmann, G.: Three-dimensional radiative transfer effects on airborne, satellite and ground-based trace gas remote sensing, Atmos. Meas. Tech., https://doi.org/10.5194/amt-2020-146, 2020.

Tack, F., Merlaud, A., Iordache, M.-D., Danckaert, T., Yu, H., Fayt, C., Meuleman, K., Deutsch, F., Fierens, F., and Van Roozendael, M.: High-resolution mapping of the $NO_2$ spatial distribution over Belgian urban areas based on airborne APEX remote sensing, Atmospheric Measurement Techniques, 10, 1665–1688, https://doi.org/10.5194/amt-10-1665-2017, https://amt.copernicus.org/articles/10/1665/2017/, 2017.

VDI - Fachbereich Umweltmeteorologie: Atmospheric dispersion models; Gaussian plume model for the determination of ambient air characteristics, Tech. Rep. VDI 3782 Blatt 1, VDI/DIN-Kommission Reinhaltung der Luft (KRdL) - Normenausschuss, 2009.

Wagner, T., Deutschmann, T., and Platt, U.: Determination of aerosol properties from MAX-DOAS observations of the Ring effect, Atmospheric Measurement Techniques, 2, 495–512, https://doi.org/10.5194/amt-2-495-2009, 2009.

Zhu, Y., Chen, J., Bi, X., Kuhlmann, G., Chan, K. L., Dietrich, F., Brunner, D., Ye, S., and Wenig, M.: Spatial and temporal representativeness of point measurements for nitrogen dioxide pollution levels in cities, Atmospheric Chemistry and Physics, 20, 13241–13251, https://doi.org/10.5194/acp-20-13241-2020, 2020.